# Para-fusion Category and Topological Defect Lines in $\mathbb{Z}_N$-parafermionic CFTs

Jin Chen[a], Babak Haghighat[b,c], and Qing-Rui Wang[c]

**a** Department of Physics, Xiamen University, Xiamen, 361005, China
**b** Yau Mathematical Sciences Center, Tsinghua University, Beijing, 100084, China
**c** Yanqi Lake Beijing Institute of Mathematical Sciences and Applications (BIMSA), Huairou District, Beijing 101408, P. R. China

September 15, 2023

## Abstract

We study topological defect lines (TDLs) in two-dimensional $\mathbb{Z}_N$-parafermoinic CFTs. Different from the bosonic case, in the 2d parafermionic CFTs, there exist parafermionic defect operators that can live on the TDLs and satisfy interesting fractional statistics. We propose a categorical description for these TDLs, dubbed as "para-fusion category", which contains various novel features, including $\mathbb{Z}_M$ $q$-type objects for $M|N$, and parafermoinic defect operators as a type of specialized 1-morphisms of the TDLs. The para-fusion category in parafermionic CFTs can be regarded as a natural generalization of the super-fusion category for the description of TDLs in 2d fermionic CFTs. We investigate these distinguishing features in para-fusion category from both a 2d pure CFT perspective, and also a 3d anyon condensation viewpoint. In the latter approach, we introduce a generalized parafermionic anyon condensation, and use it to establish a functor from the parent fusion category for TDLs in bosonic CFTs to the para-fusion category for TDLs in the parafermionized ones. At last, we provide many examples to illustrate the properties of the proposed para-fusion category, and also give a full classification for a universal para-fusion category obtained from parafermionic condensation of Tambara-Yamagami $\mathbb{Z}_N$ fusion category.

# 1  Introduction

Global symmetry has been playing a central role since the years of the foundation of quantum field theories. They provide not only guidelines in constructions of various QFTs, but also put strong constraints on the dynamics of the theories. Recently, our understanding on symmetry has been further significantly evolved [1]. In modern language, the global symmetries can be interpreted as invertible topological defects supported on codimension-1 surfaces. Such defect operators are called topological because any correlation functions with insertions of those surface operators are invariant with respect to continuous deformations of them. As a result, the topological surface operators commute with stress-energy tensor, and thus the Hamiltonian of the system as they should be. The group elements multiplication and inverse associated to the symmetry can be also re-interpreted as the fusions and orientations of the topological surfaces. With the above conceptually advanced understandings, the notion of symmetry has been greatly generalized along several directions, including higher-form/group symmetries, and non-invertible symmetries, which nowadays are called generalized global symmetries. In the former cases, all kinds of codimension-$(p + 1)$ topological surface defects can be studied in QFTs, which correspond to $p$-form symmetries, while the ordinary symmetries are of 0-form. When there are non-trivial 't Hooft anomalies among various $p$-form symmetries, they will further lead to higher group or category structures [2–8]. On the other hand, for the latter non-invertible symmetries, they are referred to as topological surface operators that are not invertible [9–30]. Therefore these topological objects are characterized by the so-called categorical symmetries that extend the notion of group symmetries by including topological surface defects with no inverse under the fusion.

    In two dimensions, the non-invertible symmetries are corresponding to 1d topological defect lines (TDLs), which are ubiquitous in 2d conformal field theories. Historically, these defect line objects were investigated because of their connections to boundary CFTs, twisted partition

functions, orbifolds, and associated SymTFTs [31–34]. However, from a modern viewpoint, the study of TDLs is providing us new tools and insights to understand various physical systems and underlying mathematical structures. For example, the generalization of 't Hooft anomalies of these TDLs helps contrain the RG flows between conformal and gapped phases [10,35–38]. The collections of non-invertible TDLs together with ordinary invertible ones, for a given 2d CFT or TFT, are described in the mathematical language of the fusion category [9,10,39–41]. Based on this language, one can also study generalized orbifolds and duality with respect to categorical symmetries. The categorical structures can be further lifted to the framework of 3d TFTs, where due to boundary-bulk correspondence, the 2d TDLs can be naturally interpreted as anyons in three dimensions [42–48]. In this picture, the study of TDLs are closely related to many interesting topics, such as the classification of topological orders, anyon condensation, and etc., in the condensed matter community [49–56].

Along this line, in recent years, the investigation of categorical symmetries has been extended to fermionic CFTs where there are fermionic degrees of freedoms satisfying Fermi-Dirac statistics [57–62]. A distinguished property of TDLs in the fermionic CFTs is that fermionic defect operators can "live" on junctions of TDLs, or "move along" some of them. With this novel feature, TDLs in fermionic systems have much richer and more involved structures from various physics and mathematical aspects. For example, a well-known example is the 2d massless Majorana theory obtained after fermionization from the Ising CFT, in which the famous non-invertible Kramers-Wannier duality $\mathcal{N}$-line in the Ising model is mapped to an *invertible but anomalous* TDL, denoted by $(-1)^{F_L}$, which is counting the left-moving fermionic number in the Majorana theory. Because of the existence of a fermionic defect operator living on $(-1)^{F_L}$, it thus admits a $\mathbb{Z}_8$-classification corresponding to $\mathrm{Hom}(\Omega_3^{\mathrm{spin}}(B\mathbb{Z}_2), \mathrm{U}(1))$ [63–66]. The mathematical structure behind TDLs in fermionic CFTs has been identified as a super-fusion category encoding extra data of fermionic defect operators [57,60,67–69]. In a parallel manner, fermionic TDLs can be lifted to 3d TFTs in terms of the so-called fermionic anyon condensation, which has received extensive attention because of its intimate relation to the classification of fermionic topological orders.

Motivated by the recent exciting progress, in this note, we aim to understand topological defect lines in two-dimensional $\mathbb{Z}_N$-parafermionic CFTs, where fermoinic TDLs can be regarded as a special case of $N = 2$. The $\mathbb{Z}_N$-parafermions, as continuous limits of lattice models, was first introduced in the seminal work of Fateev and Zemolodchikov back in the 80's of the last century [70], see also [71] for a more modern perspective. The spectra of local operators in the parafermion systems and their bosonization have been also thoroughly investigated since [36]. TDLs of the corresponding bosonized theories were recently investigated in [72]. However, to our knowledge, various novel features of TDLs in parafermionic CFTs have not been established, yet, except for the case of $N = 2$. Similar to the fermionic case, in the parafermion system, there can be exotic parafermionic defect operators living on TDLs satisfying fractional statistics. Therefore we define a new type of mathematical structure, dubbed as *para-fusion category*, to give a categorical description of these parafermionic TDLs as well as the fractional defect operators. Beside the pure 2d CFT setup, we also introduce the concept of parafermionic anyon condensation from a 3d anyon perspective. Using it, we show a type of generalized pentagon identities, dubbed as *para-pentagons*, that the parafermionic TDLs have to satisfy. To demonstrate various features of para-fusion categories, we give detailed examples in parafermionic CFTs, where the $F$-symbols of parafermionic TDLs have been solved from the proposed para-pentagons. Among these examples, we find an interesting class of para-fusion categories that we named as "para-condensed $\mathbb{Z}_N$ Tambara-Yamagami category", denoted by pf-$\mathrm{TY}_{\mathbb{Z}_N}^{t,\kappa,\beta}$. It can be obtained via parafermionic anyon condensation from the renowned $\mathbb{Z}_N$ Tambara-Yamagami category, or equivalently parafermionization of a bosonic CFT admitting $\mathbb{Z}_N$ self-duality. By solving para-pentagons, we give a full classification of pf-$\mathrm{TY}_{\mathbb{Z}_N}^{t,\kappa,\beta}$ for any $N$.

The plan of the paper is as follows: In section 2, we review the concepts of orbifolding, fermionization and parafermionization. After that, we introduce the defining properties of TDLs in parafermionic CFTs. In section 3, we lift the 2d CFTs into 3d TFT setup in terms of anyons, and introduce the concept of parafermionic anyon condensation (or para-condensation for short). Using para-condensation, we verify various important features proposed in section 2 that the parafermionic TDLs need to satisfy. In section 4 and 5, we provide many examples of TDLs in parafermionic CFTs to demonstrate our proposal. At last, in the appendix, we summarize necessary mathematics materials for the proposed $\mathbb{Z}_N$ para-fusion category.

**Note added:** When the paper is about to finish, we were informed by Zhihao Duan, Qiang Jia and Sungjay Lee that they are working on a possibly related subject [73]. We thank them for coordinating the submission.

## 2 Topological Defect Lines in Parafermionic CFTs

### 2.1 Orbifolding, Fermionization and Parafermionization

In this subsection, we will briefly review the orbifolding, fermionization and parafermionization of a given 2d bosonic CFT with a non-anomalous 0-form symmetry $G = \mathbb{Z}_N$.

**Orbifolding** It has been known for a long time that, for a given 2d CFT $\mathcal{T}$ with a non-anomalous group symmetry $G = \mathbb{Z}_N$, one can gauge this symmetry and obtain the gauged theory $\mathcal{T}' \equiv \mathcal{T}/G$, denoted as the $\mathbb{Z}_N$ orbifolded theory. More specifically, putting $\mathcal{T}$ on a torus consisting of temporal and spacial cycles, one can impose twisted boundary conditions on a local operator $\mathcal{O}(x,t)$ of $\mathcal{T}$ along the space or time direction,

$$\mathcal{O}(x+L,t) = g \cdot \mathcal{O}(x,t), \text{ or } \mathcal{O}(x,t+T) = g \cdot \mathcal{O}(x,t), \tag{1}$$

where $g \in G$ is a group element. The former boundary condition allows us to evaluate the partition function of the theory under the defect Hilbert space $\mathcal{H}_g$ twisted by $g$, while the latter one serves to project states onto different symmetry sectors with respect to $G$. Overall, for $\mathbb{Z}_N$ symmetry, the Hilbert space can be splitted into $N^2$ twisted and symmetric sectors.

In a more modern language, imposing boundary conditions onto spatial/temporal directions is equivalent to inserting different topological defect lines (TDLs) that correspond to the $\mathbb{Z}_n$ 0-form symmetries [1, 10], along the temporal/spatial directions respectively [9]. As the orbifolded one is the $G$-gauged theory $\mathcal{T}$, we can couple $\mathcal{T}$ to a background gauge field $S$, and make it dynamical, denoted as $s$. Since the to-be-gauged symmetry is a finite group, the background gauge field $S$ is necessary flat, and thus we have $S \in H^1(M, G)$, where $M$ can be a generic Riemann surface, but for our purpose, only $M = \mathbb{T}^2$ is considered throughout the paper. Rephrased in terms of topological defect lines, the background connection $S$ can be represented by a network of TDLs with trivalent junctions. When a TDL $\mathcal{L}_g$ sweeps past a local operator $\mathcal{O}$, we will have apporperiate $g$-action on the operator asin eq. (1). Therefore, placing two TDLs $\mathcal{L}_g$ and $\mathcal{L}_h$ along the spatial and temporal direction is equivalent to evaluating the following twisted partition function

$$Z_{\mathcal{T}}[g,h] = \text{Tr}_{\mathcal{H}_h} \mathcal{L}_g q^{L_0 - \frac{c}{24}} \bar{q}^{\bar{L}_0 - \frac{c}{24}}. \tag{2}$$

In this picture, it is also transparent to see how the Pontryagin dual symmetry $\hat{G}$ of $G$ emerges in the orbifold theory $\mathcal{T}'$ [9, 74]. After gauging the symmetry $G$, the corresponding

TDLs will disappear. Instead, one has Wilson lines $W_R$, corresponding to elements in $\text{Rep}(G)$, in the $G$-representation $R$ in the orbifold theory $\mathcal{T}'$, served as the TDLs therein. For $G$ non-Abelian, $\text{Rep}(G)$ is not necessary a group. However, in our setup for $G = \mathbb{Z}_N$, we have

$$\hat{G} = \text{Rep}(\mathbb{Z}_N) = \mathbb{Z}_N \tag{3}$$

in $\mathcal{T}'$. One can once again turn on a flat connection $T \in H^1(M, \hat{G})$ and couple it to the dual theory $\mathcal{T}'$, and further make it dynamical and gauge $\mathcal{T}'$ back to $\mathcal{T}$. To sum up, we have the following formula for the (twisted) partition functions of $\mathcal{T}$ and $\mathcal{T}'$,

$$Z_{\mathcal{T}'}[T] = \frac{1}{\sqrt{|H^1(M,G)|}} \sum_{s \in H^1(M,G)} e^{i(T,s)} Z_{\mathcal{T}}[s], \tag{4}$$

where $(\cdot, \cdot)$ is a bilinear map $H^1(M, \hat{G}) \times H^1(M, G) \to \mathbb{R}$, $s$ is the dynamical gauge field, and we have summed up over all twisted partition functions of $\mathcal{T}$ with respect to $s$ along different cycles of $M$ to gauge the symmetry. In the case of $M = \mathbb{T}^2$ and $G = \mathbb{Z}_N$, we make the orbifolding formula more explicit as follows,

$$Z_{\mathcal{T}'}[b_1, b_2] = \frac{1}{n} \sum_{a_1, a_2 \in \mathbb{Z}_n} \omega^{a_1 b_2 - a_2 b_1} Z_{\mathcal{T}}[a_1, a_2], \tag{5}$$

where $\omega \equiv e^{\frac{2\pi i}{N}}$ with $\gcd(p, N) = 1$, and $a_i, b_i$ corresponding to the (background) gauge fields $s$ and $T$ along the two cycles of $\mathbb{T}^2$.

**Main example.**   A primary example of such a theory with $\mathbb{Z}_N$-symmetry is the $\frac{\text{SU}(2)_N}{U(1)}$ coset CFT with central charge $c = \frac{2(N-1)}{N+2}$. The theory consists of $\frac{N(N+1)}{2}$ primaries labeled by two integers $(l, m)$ which vary in the range

$$0 \le l \le N, \quad -l + 2 \le m \le l, \quad l - m \in 2\mathbb{Z}, \tag{6}$$

with the following conformal weights

$$\Delta = \bar{\Delta} = \frac{l(l+2)}{4(N+2)} - \frac{m^2}{4N}. \tag{7}$$

The partition function in the main sector is given by

$$Z[0,0] = \sum_{l,m} \chi_{l,m}(\tau) \overline{\chi_{l,m}(\tau)}, \tag{8}$$

where the characters $\chi_{l,m}$ satisfy the relations

$$\chi_{l,m} = \chi_{l,m+2N}, \quad \chi_{l,m} = \chi_{l,-m}, \quad \chi_{l,m} = \chi_{N-l,N+m}. \tag{9}$$

The twisted sector are given by

$$Z[a_1, a_2] = e^{-2\pi i a_1 a_2 / N} \sum_{l,m} e^{2\pi i a_2 m / N} \chi_{l,m} \overline{\chi_{l,m-2a_1}}. \tag{10}$$

They satisfy the following $S$- and $T$-transformation properties

$$Z[a_1, a_2]|_T = Z[a_1, a_1 + a_2], \quad Z[a_1, a_2]|_S = Z[-a_2, a_1]. \tag{11}$$

**Fermionization**   Beside orbifolding, for a given bosonic CFT with non-anomalous $\mathbb{Z}_2 \subset \mathbb{Z}_n$, one can also fermionize the given theory. The most well-known example is the duality between the Ising model and Majorana fermion realized via the Jordan-Wigner transformation. Recently the idea of fermionization has been generalized to an arbitrary CFT with non-anomalous $\mathbb{Z}_2$ symmetry [58]. The standard procedure is to stack a Kitaev chain with a symmetry $\mathbb{Z}_{2f}$ onto the CFT, and gauge the diagonal piece of $\mathbb{Z}_2 \times \mathbb{Z}_{2f}$, i.e.

$$\mathcal{T}_f = \mathcal{T} \times \text{Kitaev}/\mathbb{Z}_2. \tag{12}$$

After fermionization, there is also an emergent $\mathbb{Z}_2$ symmetry in $\mathcal{T}_f$, denoted as $(-1)^F$ and counting the fermionic number modulo 2.

Now in $\mathcal{T}_f$, when we turn on the background gauge field $A$ for $(-1)^F$, there is a subtlety: Since the fermoinic theory will generically depend on a spin structure $\rho$ on $M$, the background gauge will shift the spin structure as $\rho \to \rho + A$. One can thus redefine the spin structure $\rho$ appropriately and absorb the background gauge field into it. Finally, the partition function of $\mathcal{T}_f$ will depend on the spin structure $\rho$, and will be related to the bosonic one as

$$Z_{\mathcal{T}_f}[\rho] = \frac{1}{\sqrt{|H^1(M,G)|}} \sum_{a \in H^1(M,\mathbb{Z}_2)} (-1)^{\text{Arf}(\rho+a)} Z_{\mathcal{T}}[a], \tag{13}$$

where the spin structure $\rho$ along a cycle of $M$ can take values of $\{0,1\}$ corresponding to the NS or Ramond sectors, respectively. In the case of $M = \mathbb{T}^2$, one can make this more explicit,

$$Z_{\mathcal{T}_f}[\rho_1, \rho_2] = \frac{1}{2} \sum_{a_1, a_2 \in \mathbb{Z}_2} (-1)^{(\rho_1+a_1)(\rho_2+a_2)} Z_{\mathcal{T}}[a_1, a_2], \tag{14}$$

or

$$Z_{\mathcal{T}}[a_1, a_2] = \frac{1}{2} \sum_{\rho_1, \rho_2 \in \mathbb{Z}_2} (-1)^{(\rho_1+a_1)(\rho_2+a_2)} Z_{\mathcal{T}_f}[\rho_1, \rho_2]. \tag{15}$$

Notice that, from eq. (13), the Arf invariant is just the non-trivial topological order of the Kitaev Majorana chain. It precisely means that our fermionization is defined by first stacking the Kitaev chain onto the bosonic theory and then gauging the diagonal $\mathbb{Z}_2$ of the two systems.

One can combine the orbifolding and fermionization operations together, so that we can first orbifold the theory $\mathcal{T}$ to $\mathcal{T}'$, and successively fermionize it to $\mathcal{T}'_f$, and thus have

$$Z_{\mathcal{T}'_f}[\rho] = \frac{1}{\sqrt{|H^1(M,G)|}} \sum_{a \in H^1(M,\mathbb{Z}_2)} (-1)^{\text{Arf}(\rho+a)+\text{Arf}(\rho)} Z_{\mathcal{T}'}[a], \tag{16}$$

or on $\mathbb{T}^2$,

$$Z_{\mathcal{T}'_f}[\rho_1, \rho_2] = \frac{1}{2} \sum_{a_1, a_2 \in \mathbb{Z}_2} (-1)^{(\rho_1+a_1)(\rho_2+a_2)+\rho_1\rho_2} Z_{\mathcal{T}'}[a_1, a_2]. \tag{17}$$

Furthermore, from eq. (13) and (16), we have the relations between $\mathcal{T}'_f$ and $\mathcal{T}_f$ as

$$Z_{\mathcal{T}'_f}[\rho] = (-1)^{\text{Arf}(\rho)} Z_{\mathcal{T}_f}[\rho], \tag{18}$$

implying the theroy $\mathcal{T}_f'$ is different from $\mathcal{T}_f$ by stacking an additional SPT phase $(-1)^{\mathrm{Arf}(\rho)}$. The above relations can be summarized in the following diagram:

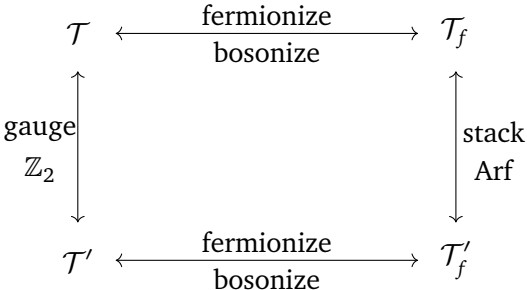

**Parafermionization**   The idea of fermionization can be further generalized to parafermionization, so long as the bosonic CFT admits a non-anomalous $\mathbb{Z}_N$ symmetry. Parallel to the discussion of fermionization where we stack a Kitaev Majorana chain onto the bosonic system, here we also need an analogous $\mathbb{Z}_N$ chain to perform the parafermionization. The $\mathbb{Z}_N$ generalization of the Ising model has been long known [70,75], where Fadeev and Zamolodchikov studied the 1d lattice model with Hamiltonian

$$H_N = -\sum_{j=1}^{L}\sum_{m=1}^{N-1} \alpha_m \left(\tau_j\right)^m - \sum_{j=1}^{L-1}\sum_{m=1}^{N-1} \beta_m \left(\sigma_j^\dagger \sigma_{j+1}\right)^m, \tag{19}$$

where $\alpha_m$ and $\beta_m$ are certain parameters, and $L$ is the number of sites of the lattice. The $N \times N$ matrices $\tau_j$ and $\sigma_j$ are generalized Pauli matrices at site $j$, satisfying

$$\sigma^N = \tau^N = 1, \;\; \sigma^\dagger = \sigma^{N-1}, \;\; \tau^\dagger = \tau^{N-1}, \;\; \text{and} \;\; \sigma\tau = \omega\tau\sigma, \tag{20}$$

with $\omega = e^{2\pi i/N}$. A representation of the above algebra can be given by diagonalizing $\tau$,

$$\tau = \mathrm{diag}\{1, \omega, \omega^2, \ldots, \omega^{N-1}\}, \;\; \sigma = \delta_{a+1,b}, \tag{21}$$

where $a, b \pmod{N}$ label the entries of the matrix $\tau$. The Hamiltionian is invariant under a $\mathbb{Z}_N$ action: $\sigma_j \to \omega\sigma_j$, and similar to the Ising chain, the corresponding symmetry generator is defined as

$$\omega^P \equiv \prod_{j=1}^{L} \tau_j^\dagger, \tag{22}$$

satisfying $\left(\omega^P\right)^N = 1$. The lattice model itself is interesting to study as it turns out to be integrable in a certain range of parameters $\alpha_m$ and $\beta_m$, as well as admits conformal phases described by the WZW coset model $\mathfrak{su}(2)_k/\mathfrak{u}(1)$. On the other hand, there is also a generalized Jordan-Wigner transformation, the Fradkin-Kadanoff transformation, that can recast the bosonic lattice model into a parafermionic chain, by defining

$$\gamma_{2j-1} \equiv \sigma_j \prod_{i<j} \tau_i, \;\; \text{and} \;\; \gamma_{2j} \equiv \omega^{(N-1)/2}\sigma_j \prod_{i\leq j} \tau_i. \tag{23}$$

With these new variables, one can show that the operators $\gamma_j$ satisfy the following statistics:

$$\gamma_j^n = 1, \;\; \gamma_j^\dagger = \gamma_j^{-1}, \;\; \text{and} \;\; \gamma_j\gamma_k = \omega^{\mathrm{sgn}(l-j)}\gamma_k\gamma_j, \;\; \text{for} \;\; j \neq k. \tag{24}$$

The last commutation relation turns out to have an interpretation from the anyon perspective that we will show in the next section. Now, analogous to the Kitaev chain, the low energy physics of the parafermionic chain is a $\mathbb{Z}_N$ topological order [76, 77]. Therefore one can similarly stack this gapped phase to a bosonic system $\mathcal{T}$ with a non-anomalous $\mathbb{Z}_N$ symmetry and gauge the diagonal group of $\mathbb{Z}_N \times \mathbb{Z}_{N,PF}$ [71, 78], i.e.

$$\mathcal{T}_{PF} = \mathcal{T} \times \text{parafermionic chain}/\mathbb{Z}_N . \tag{25}$$

The resulting theory $\mathcal{T}_{PF}$ will have an emergent $\mathbb{Z}_N$ symmetry generated by the TDL of the parafermionic chain we stacked on the original boson system.

At the level of the partition function of $\mathcal{T}_{PF}$, we can also find a para-spin structure[1] $\rho$ on $\mathbb{T}^2$, and turn on background fields $A$ for the $\mathbb{Z}_N$ in $\mathcal{T}_{PF}$. Similar to the fermionic case, we spell out the partition function of $\mathcal{T}_{PF}$ on $\mathbb{T}^2$ as

$$Z_{\mathcal{T}_{PF}}[\rho] = Z_{\mathcal{T}_{PF}}[\rho_1, \rho_2] = \frac{1}{N} \sum_{s \in H^1(\mathbb{T}^2, \mathbb{Z}_N)} \omega^{\text{Arf}_N(\rho+s)} Z_{\mathcal{T}}[s]$$

$$= \frac{1}{N} \sum_{a_1, a_2 \in \mathbb{Z}_N} \omega^{(\rho_1+a_1)(\rho_2+a_2)} Z_{\mathcal{T}}[a_1, a_2],$$

$$Z_{\mathcal{T}}[a_1, a_2] = \frac{1}{N} \sum_{\rho_1, \rho_2 \in \mathbb{Z}_N} \bar{\omega}^{(\rho_1+a_1)(\rho_2+a_2)} Z_{\mathcal{T}_{PF}}[\rho_1, \rho_2], \tag{26}$$

where $\bar{\omega} = e^{-\frac{2\pi i}{N}}$, and the para-spin structure $\rho$ along a cycle of $\mathbb{T}^2$ takes values in $\mathbb{Z}_N$ defining $N$ different sectors. For $N = 2$, it gets us back to the familiar fermionic case where we have NS and Ramond sectors. Once again, we can first orbifold $\mathcal{T}$ to $\mathcal{T}'$ and parafermionize it to $\mathcal{T}'_{PF}$, and have

$$Z_{\mathcal{T}'_{PF}}[\rho_1, \rho_2] = \frac{1}{N^2} \sum_{a_i, b_i \in \mathbb{Z}_N} \omega^{(\rho_1+b_1)(\rho_2+b_2)+a_1 b_2 - a_2 b_1} Z_{\mathcal{T}}[a_1, a_2]. \tag{27}$$

Using eq. (26), one can also establish the relation between $\mathcal{T}'_{PF}$ and $\mathcal{T}_{PF}$ [78],

$$Z_{\mathcal{T}'_{PF}}[\rho_1, \rho_2] = \frac{1}{N^3} \sum_{a_i, b_i \in \mathbb{Z}_N} \omega^{(\rho_1+b_1)(\rho_2+b_2)+a_1 b_2 - a_2 b_1 - (\tau_1+b_1)(\tau_2+b_2)} Z_{\mathcal{T}}[\tau_1, \tau_2]$$

$$= Z_{\mathcal{T}_{PF}}[\rho_1, -\rho_2] \omega^{\rho_1 \rho_2} \equiv Z_{\mathcal{T}_{PFc}}[\rho_1, \rho_2] \omega^{\rho_1 \rho_2} , \tag{28}$$

generalizing eq. (18). Overall the above relations can be also summarized as follows:

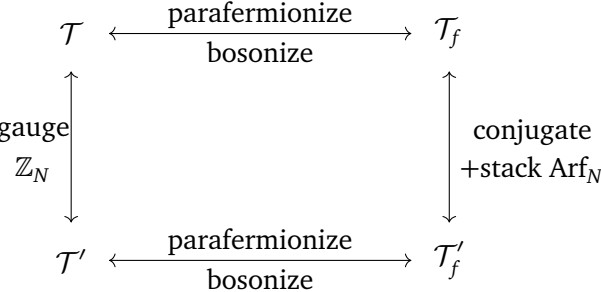

To get a better sense of parafermionization, let us now consider a simple example, the 3-Potts model as the $m = 5$ D-type minimal CFT. In terms of Virasoro algebra, the Potts model

---

[1]It is not clear so far on the existence of para-spin structures on a generic Riemann manifold.

has eight Virasoro modules with conformal weight $h = \{0, \frac{2}{5}, \frac{7}{5}, 3, \frac{1}{15}, \frac{1}{15}^*, \frac{2}{3}, \frac{2}{3}^*\}$, where the superscript "$*$" denotes that the two modules of same conformal weight are charge conjugate to each other under a charge conjugation symmetry $\mathcal{C}$. On the other hand, one can also think of the model as a diagonal CFT in terms of the $W_3$ algebra. In this picture, the primaries labeled by $(0, 3)$ and $\left(\frac{2}{5}, \frac{7}{5}\right)$ combine together and serve as irreducible modules of $W_3$. Therefore there are overall six $W_3$-modules. At the level of the partition function, this translates to

$$Z_{\mathcal{T}} = |\chi_0|^2 + |\chi_{2/5}|^2 + |\chi_{1/15}|^2 + |\chi_{1/15^*}|^2 + |\chi_{2/3}|^2 + |\chi_{2/3^*}|^2 , \tag{29}$$

where $\chi_h$ labels the character of a $W_3$-module of conformal weight $h$. Associated to these primaries, we have six Verlinde lines $\mathcal{L}_h$, and among them the lines $\{\mathcal{L}_0, \mathcal{L}_{2/3}, \mathcal{L}_{2/3^*}\}$ are corresponding to the $\mathbb{Z}_3$ symmetry of the 3-Potts. Now using eq. (27), we can obtain nine partition functions of the parafermionized theory $\mathcal{T}_{PF}$ with respect to different para-spin structures $\rho$. We here, for simplicity, only list the partition function for $\rho_i = 0$,

$$Z_{\mathcal{T}_{PF}}[0,0] = \left(\chi_0 + \chi_{2/3} + \chi_{2/3^*}\right)\bar{\chi}_0 + \left(\chi_{2/5} + \chi_{1/15} + \chi_{1/15^*}\right)\bar{\chi}_{2/5} , \tag{30}$$

from which, one can easily read off operator spectra in the $(0,0)$-sector of $\mathcal{T}_{PF}$: They are

$$\phi_{0,0}, \quad \phi_{\frac{2}{3},0}, \quad \phi^*_{\frac{2}{3},0}, \quad \phi_{\frac{2}{5},\frac{2}{5}}, \quad \phi_{\frac{1}{15},\frac{2}{5}}, \quad \phi^*_{\frac{1}{15},\frac{2}{5}} . \tag{31}$$

One can see that some of them have spin-$\frac{2}{3}$ (modulo 1), satisfying the unusual parafermionic statistics. The other partition functions with different $\rho$'s will also display similar features, and they are all related by modular transformations as summarized below following reference [71].

**S- and T-transformations.** As derived in [71], in the parafermionized theory the $S$- and $T$-transformations take the following form:

$$Z_{\mathcal{T}_{PF}}[\rho_1, \rho_2](-1/\tau) = \sum_{\rho_1', \rho_2'} S^{\rho_1', \rho_2'}_{\rho_1, \rho_2} Z_{\mathcal{T}_{PF}}[\rho_1', \rho_2'](\tau) \tag{32}$$

$$Z_{\mathcal{T}_{PF}}[\rho_1, \rho_2](\tau + 1) = \sum_{\rho_1', \rho_2'} T^{\rho_1', \rho_2'}_{\rho_1, \rho_2} Z_{\mathcal{T}_{PF}}[\rho_1', \rho_2'](\tau), \tag{33}$$

where

$$T^{\rho_1', \rho_2'}_{\rho_1, \rho_2} \equiv \frac{1}{N^2} \sum_{a_1, a_2} \omega^{(1+\rho_1+a_1)(1+\rho_2+a_2+a_1)} \overline{\omega}^{(1+\rho_1'+a_1)(1+\rho_2'+a_2)}, \tag{34}$$

$$S^{\rho_1', \rho_2'}_{\rho_1, \rho_2} \equiv \frac{1}{N^2} \sum_{a_1, a_2} \omega^{(1+\rho_1-a_2)(1+\rho_2+a_1)} \overline{\omega}^{(1+\rho_1'+a_1)(1+\rho_2'+a_2)}. \tag{35}$$

## 2.2 TDLs in parafermionic CFTs

Now we turn to characterize the properties of topological defect lines in a parafermionic CFT. Beside general features satisfied by TDLs [10], we here focus on the discussion of some unique properties particular to the TDLs in parafermionic theories. In the special case of $\mathbb{Z}_2$-parafermions, or say the usual fermion, these features have been highlighted and discussed thoroughly in [61].

### 2.2.1 Parafermion defect operators

A distinguished feature in parafermionic theories from bosonic ones is the existence of (parafermionic) zero modes. The simplest example can be seen in the fermionized Ising spin chain. After Jordan-Wigner transformation, the Ising chain is rephrased in terms of degrees of freedom of Majorana fermions which were non-local stringy operators in the original model. In the open boundary condition, one can show exactly that there are two normalized edge zero-energy modes living at the ends of the lattice chain. On the same vein, for the parafermionization of the spin chain (19), there also exist edge zero modes [76], dubbed as parafermionic defect operators, satisfying the statistics (24). Recall that parafermionic theories are obtained by stacking the $\mathbb{Z}_N$ topological order onto the corresponding boson systems. Therefore they will in general inherit these parafermionic defect operators as well.

In fermionic theories, the properties of the $\mathbb{Z}_2$-parafermionic defect operators have been discussed at length in [61]. Here we will generalize it to the parafermionic case. In the case of $\mathbb{Z}_N$-parafermionic theories, it is allowed to have parafermionic defect operators living on the junctions of TDLs. By the virtue of eq. (24), the defect Hilbert space $\mathcal{H}_{\mathcal{L}_1,\mathcal{L}_2,\mathcal{L}_3}$ associated with a 3-way junction in presence of a $\mathbb{Z}_N$-parafermionic defect operator,

$$
\begin{array}{c}
\mathcal{L}_1 \quad \mathcal{L}_2 \\
\\
\bullet\,\psi_k \\
\\
\mathcal{L}_3
\end{array}
\tag{36}
$$

is in general $N$-folded. We thus can label the TDL junctions in terms of the type of parafermionic defect operator denoted as $\psi_a$ for $a = 0, \ldots, N-1$, dubbed as "colors" of the junction. Apparently these junctions inherit the fractional statistics that a graph will pick up a phase when two of the defect operators at their junctions are switched

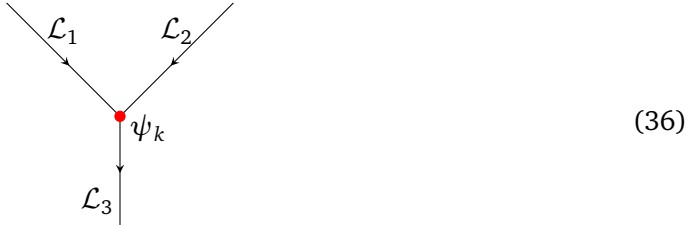

$$
\tag{37}
$$

where the phase $\theta(a, b)$ modulo unity is uniquely determined by the colors $a$ and $b$. Similarly, for the tensor product of two TDLs dressed with parafermionic defect operators, we also have

$$
\begin{array}{c}
\psi_a\,\bullet \\
\\
\\
\bullet\,\psi_b
\end{array}
\otimes
\;=\;
e^{2\pi i\theta(a,b)}
\;
\begin{array}{c}
\bullet\,\psi_b \\
\\
\\
\psi_a\,\bullet
\end{array}
\otimes
\tag{38}
$$

In the case of $N = 2$, it is known that

$$
e^{2\pi i\theta(a,b)} = (-1)^{ab}, \quad \text{for} \quad a, b = 0, 1,
\tag{39}
$$

corresponding to the fermionic statistics of the 1d Majorana zero modes when $a, b = 1$. For generic $N$, we have

$$e^{2\pi i \theta(a,b)} = \omega_{N,k}^{ab}, \quad \text{for} \quad a, b = 0, 1, \ldots, N-1, \tag{40}$$

where $\omega_{N,k} = e^{2\pi i \frac{k}{N}}$ is a primitive $N$-th root of unity with $\gcd(k, N) = 1$, which is determined by the conformal weights of the generator of $\mathbb{Z}_N$-TDLs in the original bosonic CFTs before the parafermionization. There will be a clearer explanation on (40) when, in later section, we lift the 2d story to 3d TFTs in terms of anyons, and this phase can be thus resorted to the non-trivial braidings of the corresponding $\mathbb{Z}_N$-anyons. So far, we would rather give a heuristic argument on (40): In a bosonic CFT with $\mathbb{Z}_N$-symmetries $\{\mathcal{L}_a\}_{a=0,1,\ldots,N-1}$ that could be either anomalous or non-anomalous. One can use these TDLs to prepare various defect Hilbert spaces, denoted as $\mathcal{H}_{\mathcal{L}_a}$, by inserting the TDL $\mathcal{L}_a$ along the temporal direction. In $\mathcal{H}_{\mathcal{L}_a}$, it contains defect operators $\psi_a$ with conformal weight $(h_a, 0)$. The conformal weight $h_a$ can be classified as

$$h_a = \begin{cases} \dfrac{k}{N} a^2 \mod 1, & \text{for} \quad \text{odd} \quad N \\[2mm] \dfrac{k}{2N} a^2 \mod 1, & \text{for} \quad \text{even} \quad N \end{cases}, \tag{41}$$

where $k = 0, 1, \ldots, N-1$ or $0, 1, \ldots, 2N-1$ for odd or even $N$ respectively. In the case of even $N$, the $\mathbb{Z}_N$-symmetry is *non-anomalous* only when $k$ is even, see more details in appendix A. Therefore, for non-anomalous $\mathbb{Z}_N$-symmetry with both even and odd $N$, the defect operator $\psi_a$ has conformal weight $h_a = \frac{k}{N} a^2 \mod 1$, and thus fractional spin

$$s_a = \frac{k}{N} a^2 \mod 1. \tag{42}$$

It leads to the non-trivial statistics when swapping two defect operators $\psi_a$ and $\psi_b$ as

$$\psi_a \psi_b = e^{2\pi i \sqrt{s_a s_b}} \psi_b \psi_a = e^{2\pi i \frac{k}{N} ab} \psi_b \psi_a. \tag{43}$$

For a generic $\frac{1}{N}$-fractional spin, we also require $k$ and $N$ are coprime, i.e. $\gcd(k, N) = 1$. After parafermionization, the $\mathbb{Z}_N$-TDLs are gauged away, but the defect operators $\psi_a$ are reminiscent as the 1d parafermionic defect operators living on TDLs in the parafermionic theory, and satisfy the novel statistics (40).

Clearly, due to (40), these parafermionic defect operators $\psi_a$ cannot be treated as usual local operators except for the case of $\mathbb{Z}_2$-parafermions. It can be easily seen that, by swapping two defect operators twice, there will be a phase

$$e^{4\pi i \theta(a,b)} = \omega_{N,k}^{2ab} \neq 1, \tag{44}$$

that is generically non-unity, except for $N = 2$. In this sense, the parafermionic defects cannot be regarded as local operators. In contrast, it would be necessary to always imagine that they are "stringy-like objects" that the strings attached to them are braided after swapping, and double braided for swapping twice to give the non-trivial phase in (44). In this picture, the ways, clockwise or counter-clockwise, to switch two defect operators also need to been taken into account, i.e.

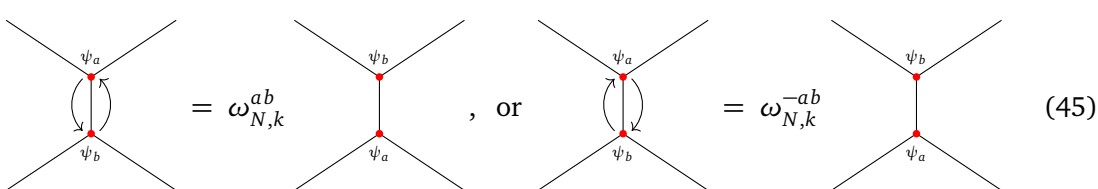

In the note, we will stick to the convention of swapping two parafermionic defect operators in the counter-clockwise fashion, and thus produce a phase as in (40). Since it is not canonical for 2d TDLs to have braiding structures, the non-locality feature of the parafermionic defect operators will be more clearly characterized when we lift the 2d theory up to 3d in terms of anyons. We will come back to this point in the next section.

### 2.2.2 $\mathbb{Z}_N$ Para-fusion category

Now we are ready to propose an axiomatic description of $\mathbb{Z}_N$ para-fusion category $\mathcal{C}_{N,p}$ for TDLs in the parafermionic CFTs. Most features of parafermionic TDLs are analogous to their bosonic cousins. On the other hand, contrary to usual fusion categories, there are two kinds of TDLs called $m$-type and $q$-type defined by parafermionic defect operators.

1. **TDLs (Objects):** An object $\mathcal{L} \in \mathcal{C}_{N,p}$ in a $\mathbb{Z}_N$ para-fusion category, is an oriented topological line operator defined on a path $C$, $\mathcal{L}(C)$, whose expectation value can be computed by inserting it in the path integral, and the dependence on $C$ is topological.

2. **Defect Operators (Morphisms):** The morphisms in $\mathcal{C}_{N,p}$ correspond to topological defect operators between two oriented lines $\mathcal{L}$ and $\mathcal{K}$. In contrast to ordinary fusion categories where the defect operators are required to be local, we here extend the notion to include the set of non-local parafermionic defect operators $\{\psi_a\}$. The defect operators form a (graded) vector space denoted by $\mathrm{Hom}(\mathcal{L}, \mathcal{K})$. In addition, given defect operators $\mu \in \mathrm{Hom}(\mathcal{L}, \mathcal{K})$ and $\nu \in \mathrm{Hom}(\mathcal{K}, \mathcal{J})$, there is a composition of defect operators $\mu$ and $\nu$, denoted as $\nu \circ \mu \in \mathrm{Hom}(\mathcal{L}, \mathcal{J})$, that can be thought to shrink the segment of topological line $\mathcal{K}$ to bring the two defect operator $\mu$ and $\nu$ together. Especially, for the composition of two parafermionic defect operators, we have

$$\psi_a \circ \psi_b = \psi_{a+b \bmod N}. \tag{46}$$

3. **Additive Structure:** Given two TDLs $\mathcal{L}$ and $\mathcal{K}$, we define a new TDL as the sum of $\mathcal{L}$ and $\mathcal{K}$, denoted by $\mathcal{L} \oplus \mathcal{K}$, such that

$$\langle \cdots (\mathcal{L} \oplus \mathcal{K})(C) \cdots \rangle = \langle \cdots \mathcal{L}(C) \cdots \rangle + \langle \cdots \mathcal{K}(C) \cdots \rangle. \tag{47}$$

4. **Simplicity, Semisimplicity, and Finiteness:** A simple TDL in a para-fusion category $\mathcal{C}_{N,p}$ is defined as lines that cannot be decomposed as a sum of other TDLs. Further, $\mathcal{C}_{N,p}$ is semisimple if any TDL in it is isomorphic to a sum of simple lines in it. At last, we require finiteness of $\mathcal{C}_{N,p}$ in the sense that the number of simple lines in $\mathcal{C}_{N,p}$ is finite.

5. **$m$-type and $q$-type of TDLs:** A simple $m$-type TDL $\mathcal{L}_m$ is defined as a simple line whose defect space $\mathrm{Hom}(\mathcal{L}_m, \mathcal{L}_m)$ only contains the trivial parafermionic defect $\psi_0$, and thus we have

$$\mathrm{Hom}(\mathcal{L}_m, \mathcal{L}_m) = \mathrm{Span}\{\psi_0\} \simeq \mathbb{C}^{\overbrace{1|0|\cdots|0}^{N}}, \tag{48}$$

where the Hom space is $N$-graded and the subscription of $\mathbb{C}$ labels the type of parafermionic defect operators, dubbed as "colors". At last, a simple $q$-type TDL $\mathcal{L}_q$ is defined as a simple line whose defect space $\mathrm{Hom}(\mathcal{L}_q, \mathcal{L}_q)$ contains at least a non-trivial parafermionic defect $\psi_a$ for $a \neq 0$. An interesting feature of the $q$-type TDLs can be established from (46): Given a parafermionic defect $\psi_{a\neq 0} \in \mathrm{Hom}(\mathcal{L}_q, \mathcal{L}_q)$, consider the TDL $\mathcal{L}_q$ dressed

with two such $\psi_a$ defect operators. One can shrink the segment of $\mathcal{L}_q$ between the two $\psi_a$'s and bring them together. Using (46), one can show that

$$\psi_a \circ \psi_a = \psi_{2a \bmod N} \in \mathrm{Hom}(\mathcal{L}_q, \mathcal{L}_q). \tag{49}$$

Repeating this procedure, we can find a collection of $\{\psi_a\}$, with $a \in \mathbb{Z}_M$ and $M$ dividing $N$, in $\mathrm{Hom}(\mathcal{L}_q, \mathcal{L}_q)$. In general, for $\mathrm{Hom}(\mathcal{L}_q, \mathcal{L}_q)$ spanned by a collection of $S = \{\psi_a\}$, one can always find a smallest $\psi_M$ of order $M$ generating the whole set $S = \langle \psi_M \rangle$. We thus denote such a TDL as a "$\mathbb{Z}_M$ $q$-type TDL". For example, for a $\mathbb{Z}_N$ $q$-type TDL $\mathcal{L}_q$, its Hom space is given by

$$\mathrm{Hom}(\mathcal{L}_q, \mathcal{L}_q) = \mathrm{Span}\{\psi_1\} \simeq \mathbb{C}^{\overbrace{1|\cdots|1}^{N}}. \tag{50}$$

6. **Fusion:** Given two TDLs $\mathcal{J}$ and $\mathcal{K}$, one can bring them close enough to fuse them as a single line, denoted by $\mathcal{J} \otimes \mathcal{K}$. By semisimplicity, $\mathcal{J} \otimes \mathcal{K}$ can be decomposed as a sum of simple TDLs $\{\mathcal{L}_i\}$, which is determined by the Hom space, $\mathrm{Hom}(\mathcal{J} \otimes \mathcal{K}, \mathcal{L})$. Obviously parafermionic defect operators $\{\psi_a\}$ in $\mathrm{Hom}(\mathcal{J} \otimes \mathcal{K}, \mathcal{L})$ define the colors of the 3-way junction.

7. **Associativity Structure:** The fusion operation is associative in the sense that $(\mathcal{I} \otimes \mathcal{J}) \otimes \mathcal{K}$ and $\mathcal{I} \otimes (\mathcal{J} \otimes \mathcal{K})$ are isomorphic by an associator

$$\mathcal{F}^{\mathcal{I}, \mathcal{J}, \mathcal{K}} \in \mathrm{Hom}\left((\mathcal{I} \otimes \mathcal{J}) \otimes \mathcal{K}, \mathcal{I} \otimes (\mathcal{J} \otimes \mathcal{K})\right), \tag{51}$$

where the associator $\mathcal{F}$ is also called $F$-symbol. In the para-fusion category $\mathcal{C}_{N,p}$, it is colored because of the four 3-way junctions in $\mathrm{Hom}\left((\mathcal{I} \otimes \mathcal{J}) \otimes \mathcal{K}, \mathcal{I} \otimes (\mathcal{J} \otimes \mathcal{K})\right)$. Expanded in a base $S = \{\mathcal{L}_i\}$ of simple lines, we have

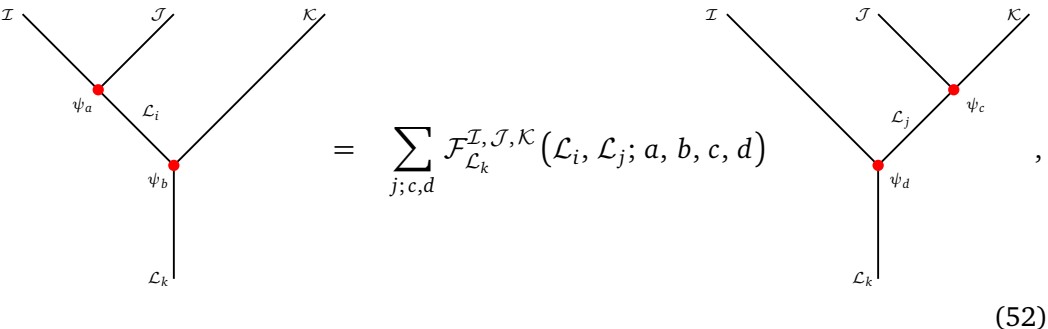

$$\tag{52}$$

where we have used $\{\mathcal{L}_i\}$ to decompose the fusions of $\mathcal{I} \otimes \mathcal{J}$, $\mathcal{L}_i \otimes \mathcal{K}$, $\mathcal{J} \otimes \mathcal{K}$, and $\mathcal{I} \otimes \mathcal{L}_j$. In addition, there is a selection rule imposed on the colors of a given $F$-symbol element because of the emergent $\mathbb{Z}_N$-symmetry (22), that

$$\mathcal{F}^{\mathcal{I}, \mathcal{J}, \mathcal{K}}_{\mathcal{L}_k}\left(\mathcal{L}_i, \mathcal{L}_j; a, b, c, d\right) = 0, \qquad \text{if} \qquad a + b \neq c + d \mod N. \tag{53}$$

8. **Para-fusion Pentagons:** As in a usual fusion category, $F$-symbols in para-fusion categories need to satisfy pentagon identities. The novel feature here is that there are different colors in each 3-way junction. When swapping the positions of these junctions, there are extra phases produced due to (40). In the case of $\mathbb{Z}_2$-parafermions, the phase is simply "$-1$", and the pentagons are modified to super-pentagons [57, 65]. We here

generalize it to arbitrary $\mathbb{Z}_N$-parafermions. Consider the following diagrams with five external TDLs,

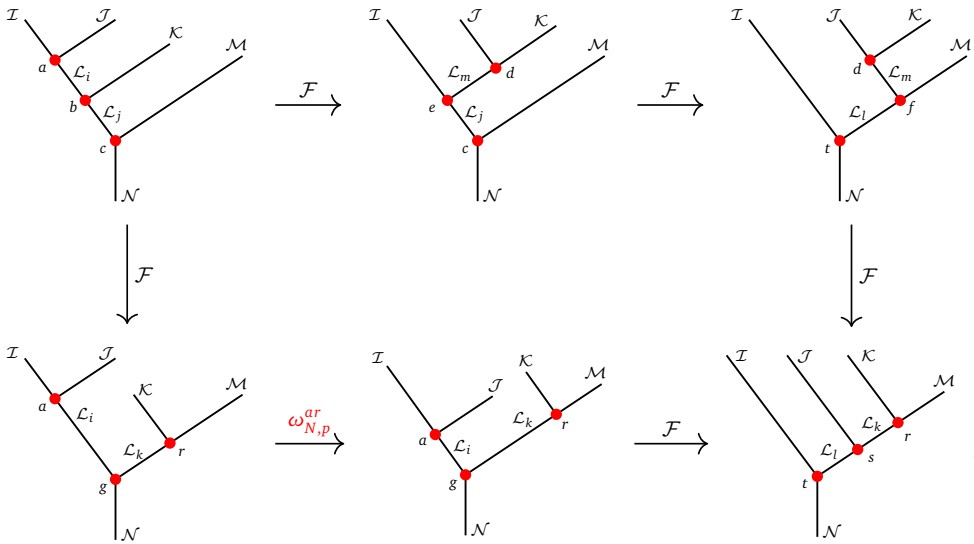

where, when switching the position of colors "$a$" and "$r$" from the first to the second diagram of the second line, we have employed (37) and (40) to pick up a phase $\omega_{N,p}^{ar}$. The two routes, to bring the above diagrams from left top corner to the right bottom one, give us constraints on the $F$-symbols, denoted as the para-pentagon equations in $\mathcal{C}_{N,p}$. In the base $S = \{\mathcal{L}_i\}$, the para-pentagons of above diagrams spell as

$$\sum_{m;d,e,f} \mathcal{F}_{\mathcal{L}_j}^{\mathcal{I},\mathcal{J},\mathcal{K}}(\mathcal{L}_i,\mathcal{L}_m;a,b,d,e) \, \mathcal{F}_{\mathcal{N}}^{\mathcal{I},\mathcal{L}_m,\mathcal{M}}(\mathcal{L}_j,\mathcal{L}_l;e,c,f,t) \, \mathcal{F}_{\mathcal{L}_l}^{\mathcal{J},\mathcal{K},\mathcal{M}}(\mathcal{L}_m,\mathcal{L}_k;d,f,r,s)$$

$$= \sum_g \omega_{N,p}^{ar} \, \mathcal{F}_{\mathcal{N}}^{\mathcal{L}_i,\mathcal{K},\mathcal{M}}(\mathcal{L}_j,\mathcal{L}_k;b,c,r,g) \, \mathcal{F}_{\mathcal{N}}^{\mathcal{I},\mathcal{J},\mathcal{L}_k}(\mathcal{L}_i,\mathcal{L}_l;a,g,s,t) \,, \tag{54}$$

where we have projected the para-pentagon equations to the diagram with internal TDLs $(\mathcal{L}_k, \mathcal{L}_l)$, and colors $(r, s, t)$.

## 3   Categorical Parafermionic Anyon Condensation

In previous sections, we discussed parafermionization in 2d CFTs. However, parafermionization can also be applied to 3d models, which can be viewed as the bulk of the 2d CFTs. By incorporating a 3d bulk model into our analysis, we gain new insights into the behavior of parafermionic anyons in 3d and their relation to parafermionization in 2d. This leads to a more complete understanding of the underlying mechanisms at play, providing valuable insights into the behavior of the 3d bulk topological order and its 2d boundary in general.

In this section, we will explore how parafermionic anyon condensation in a 3d bulk Turaev-Viro-Levin-Wen string-net model [79,80] is related to the parafermionization of the 2d boundary theory. Parafermionic anyon condensation provides us with a categorical understanding of the parafermionization, allowing us to better understand the relation of TDLs of the models before and after parafermionization.

The main result is that, if the TDLs in the original 2d theory are described by a fusion category $\mathcal{C}$, then after the bosonic/fermionic/parafermionic anyon condensation of the algebra

$A$ under several conditions, the TDLs of the condensed theory will be a bimodule category ${}_A\mathcal{C}_A$, which is a fusion/super-fusion/para-fusion category. From the point of view of 3d topological orders, the generalized Turaev-Viro-Levin-Wen string-net model before and after condensation are constructed with the input fusion category $\mathcal{C}$ and fusion/super-fusion/para-fusion category ${}_A\mathcal{C}_A$ or $\mathcal{C}_A$, respectively. There is a gapped domain wall between these two topological orders.

## 3.1 Bosonic anyon condensation

Bosonic anyon condensation was first discussed in the mathematical community [81, 82] and was later independently discovered as a phenomenon that can occur in 3d topological orders [53–56]. In 3d topological orders, spatial point-like topological excitations, called anyons, can fuse and braid with each other. The process of fusing and braiding anyons can be mathematically described using a unitary modular tensor category (UMTC) framework [83]. Anyon condensation is then a procedure in which a new UMTC is obtained from an old one by condensing some bosonic anyons. It is also recognized that the condensation of bosonic anyons in 3d topological order is intimately connected to the conformal extension of 2d CFTs [53].

Apart from considering anyon condensation as a phase transition from one topological order to another, one can also view it as a gapped domain wall separating two distinct topological orders [84, 85]. As a special case, the gapped boundary of a topological order can be seen as the domain wall between the topological order and the vacuum state. In this way, we can understand both the domain wall and the boundary on equal terms.

In fact, we can differentiate between two types of bosonic anyon condensations, which will be discussed separately in the following. The first type is more commonly discussed in the condensed matter physics literature. It occurs in a UMTC $\mathcal{B}$, where both the fusion and braiding of the anyons are defined. The second type, which will be more focused on in this paper when discussing TDLs, occurs in the unitary fusion category (UFC) $\mathcal{C}$ where there are no braiding operations in general. In the following, we will use the notation $\mathcal{B}$ for a UMTC to indicate that it is braided. On the contrary, a fusion category without braiding will be denoted as $\mathcal{C}$.

### 3.1.1 Bosonic anyon condensation in UMTC $\mathcal{B}$

In a quantum field theory, boson condensation or Bose-Einstein condensation occurs when a boson $b$ is condensed by changing its effective mass to negative. In the condensed phase, the boson $b$ can be created or annihilated arbitrarily in the new vacuum. For this new vacuum to be consistent, the self-braiding of $b$ must be trivial, meaning that $b$ should be a boson. Otherwise, creating, braiding, and annihilating a pair of $b$ will result in a nontrivial phase factor that is ambiguous for the vacuum.

The concept of boson condensation is generalized to 3d topological orders or anyon models described by a unitary modular tensor category (UMTC) $\mathcal{B}$ [53–56, 81]. Given a subset $A$ of anyons in $\mathcal{B}$, we can perform anyon condensation by identifying $A$ as the new vacuum in the condensed phase. To ensure the consistency of this identification, certain technical conditions must be satisfied such that $A$ forms a connected commutative separable algebra in $\mathcal{B}$ [55, 81]. For example, if we intend to condense a single Abelian boson $b$ with $\mathbb{Z}_2$ fusion rule, the algebra $A$ could be represented as $1 \oplus b$.

When we perform an anyon condensation of $A$ by identifying $A$ as the new vacuum, we obtain an intermediate phase described by a fusion category $\mathcal{B}_A$ (without braidings in general). The fusion category $\mathcal{B}_A$ consistes of $A$-modules and module morphisms. The total quantum

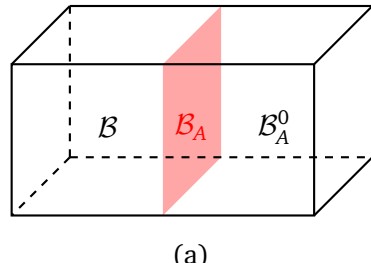

(a)

Figure 1: Bosonic anyon condensation in 3d topological order of UMBC $\mathcal{B}$. $A$ is a connected commutative separable algebra in $\mathcal{B}$. The red domain wall that separates the topological orders before and after the condensation is described by the fusion category $\mathcal{B}_A$ of the right $A$-modules. Subsequently, the condensed phase is characterized by the UMTC $\mathcal{B}_A^0$ which consists of local (or dyslectic) right $A$-modules in $\mathcal{B}$.

dimension of $\mathcal{B}_A$ is related to that of $\mathcal{B}$ by

$$\dim(\mathcal{B}_A) := \sum_{a \in \mathcal{B}_A} d_a^2 = \frac{\dim \mathcal{B}}{\dim A}. \tag{55}$$

Physically, this intermediate phase can be considered as a 2d *gapped domain wall* between the original and final 3d phases of the condensation procedure. The objects in $\mathcal{B}_A$ correspond to the excitations localized at the domain wall.

The condensed phase resulting from anyon condensation by $A$ gives rise to a 3d topological order that can be described by a UMTC $\mathcal{B}_A^0$. Unlike the intermediate fusion category phase $\mathcal{B}_A$, $\mathcal{B}_A^0$ possesses braidings as part of its structure. To obtain $\mathcal{B}_A^0$, we select the local or dyslectic $A$-modules [86] from $\mathcal{B}_A$. The key principle is that anyons in $\mathcal{B}_A^0$ should have trivial full braidings with $A$ since $A$ is new vacuum. Any anyons that exhibit nontrivial full braidings with $A$ become confined in the condensed phase due to the quantum interference of Aharonov-Bohm effects. It can be shown that the total quantum dimension of $\mathcal{B}_A^0$ is given by

$$\dim(\mathcal{B}_A^0) = \frac{\dim(\mathcal{B}_A)}{\dim A} = \frac{\dim \mathcal{B}}{(\dim A)^2}. \tag{56}$$

### 3.1.2 Bosonic anyon condensation in UFC $\mathcal{C}$

Anyon condensation can be defined not only in braided tensor category $\mathcal{B}$ but also in fusion category $\mathcal{C}$. In this case we will obtain another fusion category from a given fusion category $\mathcal{C}$ by condensing an algebra $A$ with some conditions.

In 2d theory, anyon condensation of an algebra $A$ in $\mathcal{C}$ has a physical interpretation as a generalized notion of gauging [9]. Suppose we have a 2d theory with fusion category symmetry $\mathcal{C}$, which means that the TDLs of the theory are labeled by objects in $\mathcal{C}$. Gauging an algebra $A$ in $\mathcal{C}$ involves labeling all lines of a fine enough mesh in 2d spacetime with $A$. In the subsequent discussion, it becomes clear that the resulting theory exhibits an emergent fusion category symmetry ${}_A\mathcal{C}_A$, depicted on the upper surface of Figure 2(a).

In the context of 3D bulk theory, the physical interpretation of anyon condensation within a fusion category becomes most clear when considering the topological order of Levin-Wen string-net models [80]. The string-net model is essentially a Hamiltonian version on a 2D spatial manifold that corresponds to the Turaev-Viro state sum construction used for calculating the 3D partition function [79]. The input data for the Hamiltonian is a fusion category $\mathcal{C}$, and the output topological order is a 3d TQFT with UMTC $Z(\mathcal{C})$ of Reshetikhin-Turaev type [87]. The process of condensation enables the creation of a gapped domain wall that separates two

distinct Levin-Wen models, one before the condensation and one after. Depending on the details of the algebra $A$, the input category for the Hamiltonian of the condensed Levin-Wen model can be either $_A\mathcal{C}_A$ or $_A\mathcal{C}$ [see Figures 2(a) and 2(b)]. In the following, we will discuss these two types of condensation in a fusion category $\mathcal{C}$ separately.

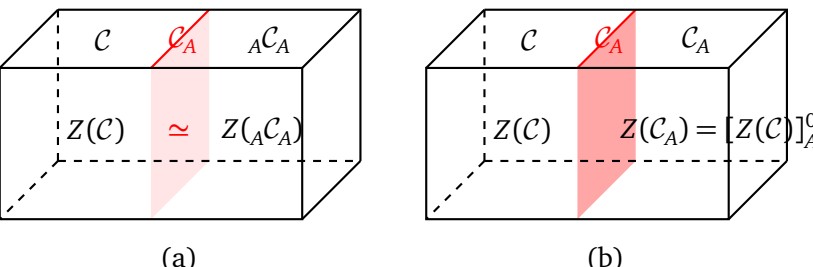

(a)  (b)

Figure 2: Two types of bosonic anyon condensation in UFC $\mathcal{C}$ corresponding to a 2d boundary or UMTC $Z(\mathcal{C})$ associated with a 3d bulk topological order. The 2d boundary can be understood from two different perspectives: as a 2d spatial manifold used to define the Hamiltonian, or as a 1+1d boundary through a Wick rotation. (a) $A$ is an algebra in fusion category $\mathcal{C}$. The domain wall $\mathcal{C}_A$ has no monoidal/tensor structure in general. The original fusion category $\mathcal{C}$ and the bimodule category $_A\mathcal{C}_A$ are Morita equivalent, i.e., $Z(\mathcal{C}) \simeq Z(_A\mathcal{C}_A)$. So the red domain wall between them is invertible or an equivalence of bulk UMTC. A prototypical example is $\mathcal{C} = \text{Vec}_G$, $A = \mathbb{C}[G]$, $\mathcal{C}_A = \text{Vec}$ and $_A\mathcal{C}_A = \text{Rep}(G)$, where the two sides are related by gauging/condensing $A$. (b) $A$ is a central commutative algebra, i.e., a commutative algebra in $Z(\mathcal{C})$. The category of right-$A$ modules already has a monoidal/tensor structure. The bulk of $\mathcal{C}$ and $\mathcal{C}_A$ are related by $Z(\mathcal{C}_A) = [Z(\mathcal{C})]_A^0$ [88]. So the 3d red domain wall corresponds to anyon condensation of topological orders from UMTC $Z(\mathcal{C})$ to $[Z(\mathcal{C})]_A^0$.

**(a) $A$ is an algebra of $\mathcal{C}$ [see Figure 2(a)]**

Let's begin by assuming that $A$ is an algebra within the fusion category $\mathcal{C}$, which characterizes the TDLs of a 2D theory, whether it's a gapped system or a gapless one like a CFT. Physically, condensing $A$ means gauging or orbifolding $A$ in the original 2d theory [9]. To be more precise, gauging or condensing $A$ means summing over all configurations of TDLs labeled by $A$ on the 2d spacetime. Therefore, the final 2D theory is defined on a 2D spacetime with a fine enough mesh, where all lines are labeled by $A$.

We can inquire about the fusion category symmetry exhibited by the condensed theory. The TDLs of the condensed theory are lines that can absorb or emit $A$ lines from both the left and right sides. Mathematically, they form the fusion category $_A\mathcal{C}_A$ of $A$-bimodules. It is fusion because we can define the tensor structure $\otimes_A$ over $A$. On the domain wall separating the two theories, the $A$ lines can only merge from the right-hand side. Therefore, the 1d domain wall is labeled by objects in the category (not necessarily a fusion category) $\mathcal{C}_A$ of right $A$-modules. In fact, $\mathcal{C}_A$ is a $(\mathcal{C}, _A\mathcal{C}_A)$-bimodule category since TDLs in $\mathcal{C}$ and $_A\mathcal{C}_A$ can fuse to it from the left and right-hand sides, respectively.

Let us delve into more details about the $A$-module category $\mathcal{C}_A$ [41]. For any right $A$-module $M \in \mathcal{C}_A$, there exists an object $X \in \mathcal{C}$ and a surjection $X \otimes A \to M$. Consequently, any irreducible right $A$-module in $\mathcal{C}_A$ is a quotient of a module of the form $X \otimes A$. The object $X \otimes A$ possesses a natural right $A$-module structure given by the composition of the associator in $\mathcal{C}$ and the multiplication of $A$: $(X \otimes A) \otimes A \to X \otimes (A \otimes A) \to X \otimes A$. Therefore, we have a $\mathcal{C}$-module functor $-\otimes A : \mathcal{C} \to \mathcal{C}_A, X \mapsto X \otimes A$. There is another $\mathcal{C}$-module functor $Forg : \mathcal{C}_A \to \mathcal{C}$ that forgets the $A$-module structure of an object. In fact, these two functors are adjoint to each other, meaning

that the Hom spaces are related as follows:

$$\text{Hom}_{\mathcal{C}_A}(X \otimes A, M) = \text{Hom}_{\mathcal{C}}(X, Forg(M)). \tag{57}$$

Sometimes, we abuse the notation by letting $M$ denote $Forg(M) \in \mathcal{C}$. It is a consequence of the special case that $\text{Hom}_{\mathcal{C}_A}(A, M) = \text{Hom}_{\mathcal{C}}(1, M)$ for any $M$ in $\mathcal{C}_A$.

The adjoint relation of the two functors helps us understand the relations of objects in $\mathcal{C}_A$ in terms of those in $\mathcal{C}$. Let's take the example of $A = 1 \oplus b$ to illustrate this. We have:

$$\begin{aligned}
\text{Hom}_{\mathcal{C}_A}(X \otimes A, Y \otimes A) &= \text{Hom}_{\mathcal{C}}(X, Y \otimes A) \\
&= \text{Hom}_{\mathcal{C}}(X, Y) \oplus \text{Hom}_{\mathcal{C}}(X, Y \otimes b).
\end{aligned} \tag{58}$$

We see that the Hom space of $\mathcal{C}_A$ after the $A = 1 \oplus b$ condensation is a direct sum of two Hom spaces from the original category $\mathcal{C}$. The physical interpretation is that we do not distinguish between $Y$ and $Y \otimes b$ after the condensation, as $b$ is considered to be in the new vacuum. If we set $X = Y$, we can also use it to calculate

$$\text{End}_{\mathcal{C}_A}(X \otimes A) := \text{Hom}_{\mathcal{C}_A}(X \otimes A, X \otimes A) = \text{Hom}_{\mathcal{C}}(X, X \otimes A). \tag{59}$$

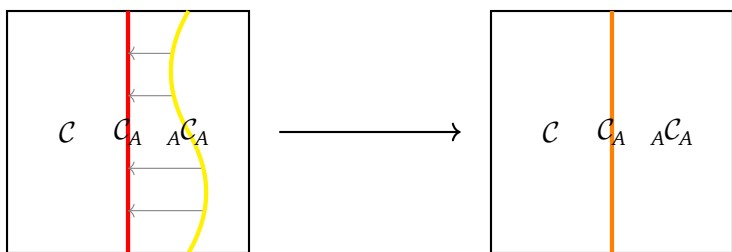

Figure 3: Physical interpretation of the fusion category equivalence ${}_A\mathcal{C}_A \simeq \text{Fun}_{\mathcal{C}}(\mathcal{C}_A, \mathcal{C}_A)$. The fusion category $\mathcal{C}$ and ${}_A\mathcal{C}_A$ describe the TDLs of the left original theory and the right $A$-condensed theory in 2d, respectively. They are separated by a (red) domain wall in the middle. The domain wall is labeled by objects in the module category $\mathcal{C}_A$. When moving a (yellow) TDL $M \in {}_A\mathcal{C}_A$ from the right condensed theory to the middle (red) domain wall labeled by object in $\mathcal{C}_A$, the domain wall is changed to another (orange) one in $\mathcal{C}_A$. This process of fusing $M \in {}_A\mathcal{C}_A$ provides a functor from $\mathcal{C}_A$ to itself, establishing an equivalence between the TDLs in ${}_A\mathcal{C}_A$ and the boundary condition changing processes in $\text{Fun}_{\mathcal{C}}(\mathcal{C}_A, \mathcal{C}_A)$ from $\mathcal{C}_A$ to itself.

As previously mentioned, the TDLs of the condensed theory are described by the fusion category symmetry ${}_A\mathcal{C}_A$. In terms of right module category, this bimodule category can also be understood as the right $A^{\text{op}} \otimes A$-module, where $A^{\text{op}}$ is $A$ with the opposite tensor product. Another way to understand this bimodule category is through module categories, which is represented by the monoidal equivalence [41]

$$_A\mathcal{C}_A \simeq \text{Fun}_{\mathcal{C}}(\mathcal{C}_A, \mathcal{C}_A). \tag{60}$$

Here, $\text{Fun}_{\mathcal{C}}(\mathcal{C}_A, \mathcal{C}_A)$ is the category of $\mathcal{C}$-module functors between $\mathcal{C}_A$ and $\mathcal{C}_A$ with certain technical conditions. The tensor structure of this category comes from the composition of functors. In this case, the equivalence between the two fusion categories is established by mapping an $A$-bimodule $M$ to $- \otimes_A M$, forming a functor from $\mathcal{C}_A$ to $\mathcal{C}_A$. This mapping has a physical interpretation as moving a TDL in the condensed theory ${}_A\mathcal{C}_A$ to the domain wall labeled by $\mathcal{C}_A$,

resulting in another type of domain wall. The equivalence $_A\mathcal{C}_A \simeq \text{Fun}_\mathcal{C}(\mathcal{C}_A, \mathcal{C}_A)$ indicates that all domain wall-changing processes can be obtained in this manner (see Figure 3).

Another way to relate the $A$-bimodule category and the left/right $A$-module category is through the equivalence [89]

$$_A\mathcal{C}_A \simeq (_A\mathcal{C}) \boxtimes_\mathcal{C} (\mathcal{C}_A). \tag{61}$$

Here, the right-hand side involves the relative Deligne tensor product of two module categories over $\mathcal{C}$. The physical meaning of this relation can be understood as follows. Consider the original 2d theory with the geometry of a stripe and TDLs labeled by $\mathcal{C}$. Let us condense $A$ to obtain a new theory with TDLs in $_A\mathcal{C}_A$ from both the left and right sides. As a result, there are two domain walls, $_A\mathcal{C}$ and $\mathcal{C}_A$, separating three 2d theories: $_A\mathcal{C}_A$, $\mathcal{C}$, and $_A\mathcal{C}_A$. By shrinking the $\mathcal{C}$ stripe or fusing the two domain walls, we obtain a 1d topological line that lives in the 2d theory $_A\mathcal{C}_A$. This line should also be labeled by objects in $_A\mathcal{C}_A$. The fusion process is mathematically represented by the relative Deligne tensor product $(_A\mathcal{C}) \boxtimes_\mathcal{C} (\mathcal{C}_A)$. Consequently, we find the equivalence of the two fusion categories $(_A\mathcal{C}) \boxtimes_\mathcal{C} (\mathcal{C}_A)$ and $_A\mathcal{C}_A$.

The standard example of such anyon condensation in a fusion category is the case of $\mathcal{C} = \text{Vec}_G$, where $G$ is a finite group [9]. The algebra to be condensed is the group ring $A = \mathbb{C}[G]$. The process of condensing the algebra $A$ can be understood as physically gauging the symmetry group $G$. This interpretation arises because summing over all possible flat connections of $G$ is equivalent to introducing additional $A$ lines into the space. In this context, the resulting gapped domain wall $\mathcal{C}_A$ is simply $\mathcal{C}_A = \text{Vec}$, indicating that there is only one nontrivial simple excitation on the domain wall. The gauged phase corresponds to $_A\mathcal{C}_A = \text{Rep}(G)$, which represents the category of $G$ representations. It's worth noting that it is widely recognized that $Z(\text{Vec}_G) \simeq Z(\text{Rep}(G))$. This equivalence establishes that the two fusion categories $\text{Vec}_G$ and $\text{Rep}(G)$ are Morita equivalent.

**(b) $A$ is a central commutative algebra of $\mathcal{C}$ [see Figure 2(b)]**

Mathematically, the condensed subset $A$ is an algebra (without braidings) within the fusion category $\mathcal{C}$, not an "anyon" as anyons in 3D exhibit braidings. However, we term it "anyon" as our primary focus is on situations where $A$ is a central commutative algebra within $\mathcal{C}$. This means that we can lift $A$ to be an object in $Z(\mathcal{C})$, and ensure that $A$ maintains commutativity within $Z(\mathcal{C})$. In this way, $A$ in fact has the braiding structure with all objects in $\mathcal{C}$. It is also this case when considering generating it to fermionic or parafermionic cases later.

Let's now assume that $A$ is a central commutative algebra within $\mathcal{C}$. In a similar manner as before, condensing $A$ gives rise to a domain wall consisting of right-$A$ modules, collectively forming a category denoted as $\mathcal{C}_A$. The key aspect here is that the requirement for $A$ to be a commutative algebra within $Z(\mathcal{C})$ enables us to braid $A$ from the left side of an object $X \in \mathcal{C}_A$ to its right side, subsequently acting on $X$ from the right. As a result, a right-$A$ module automatically becomes a left-$A$ module as well.

Consequently, $\mathcal{C}_A$ can be understood as a subcategory of $_A\mathcal{C}_A$, where the monoidal/tensor structure is naturally present within $\mathcal{C}_A$. This observation implies that, following the condensation, the original fusion category $\mathcal{C}$ transforms into a theory exhibiting a fusion category (sub-)symmetry represented by $\mathcal{C}_A$.

### 3.1.3 Relation between condensations in $\mathcal{C}$ and $\mathcal{B} = Z(\mathcal{C})$

Previously, we approached the topic of bosonic anyon condensation within $\mathcal{C}$ from a 2d theory viewpoint. Alternatively, it is also possible to investigate anyon condensation from a 3d bulk theory perspective by employing Levin-Wen string-net models [as illustrated in Figure 2].

Rather than contemplating a 1+1d CFT boundary, we can apply a Wick rotation to consider the boundary as a 2d spatial manifold. In this setup, the fusion category $\mathcal{C}$ governing the TDLs

within the boundary serves as the input data for constructing the Levin-Wen Hamiltonians. The anyonic excitations present in the 3d topological order can then be described through the Drinfeld center $Z(\mathcal{C})$.

The condensation process of $A$ in $\mathcal{C}$ on the 2d boundary can also be related to anyon condensation within the underlying UMTC $Z(\mathcal{C})$ in the 3d bulk. In the subsequent discussions, we will delve into an exploration of the two distinct types of $A$ condensation separately.

**(a) $A$ is an algebra of $\mathcal{C}$ [see Figure 2(a)]**

For a given input fusion category $\mathcal{C}$, its objects are used to label the string types within the Levin-Wen string-net model. In the condensed model, the string types are labeled by objects from $_A\mathcal{C}_A$. In the context of the 3d bulk view, the anyonic excitations on the left and right sides correspond to Drinfeld centers $Z(\mathcal{C})$ and $Z(_A\mathcal{C}_A)$, respectively. These two sides are separated by a gapped domain wall, depicted as the red line in Figure 2(a).

A natural question emerges: what is the relationship between the anyons in the left and right Levin-Wen models? An important mathematical result establishes the equivalence between these two UMTCs:

$$Z(\mathcal{C}) \simeq Z(_A\mathcal{C}_A), \tag{62}$$

known as Morita equivalence. Consequently, the red domain wall surface depicted in Figure 2 is invertible. It represents a self-duality wall within the same 3D topological order.

**(b) $A$ is a central commutative algebra of $\mathcal{C}$ [see Figure 2(b)]**

When considering the scenario where $A$ is a central commutative algebra within $\mathcal{C}$, the fusion category on the right-hand side can be selected as $\mathcal{C}_A$. We now have two Levin-Wen string-net models with input fusion categories $\mathcal{C}$ and $\mathcal{C}_A$. In this situation, the topological orders described by $Z(\mathcal{C})$ and $Z(\mathcal{C}_A)$ on the two sides are not equivalent. Instead, the latter can be interconnected through the anyon condensation process within the initial UMTC $Z(\mathcal{C})$. Schauenburg's theorem [88] reveals that

$$Z(\mathcal{C}_A) \simeq [Z(\mathcal{C})]_A^0. \tag{63}$$

Here, the $A$ on the right-hand side is treated as a commutative algebra within $Z(\mathcal{C})$, thereby allowing us to perform anyon condensation within this UMTC.

In fact, lattice models for bosonic anyon condensation have also been formulated [90, 91]. One can explicitly demonstrate that the complete Hamiltonian, including those on both sides as well as the one on the intermediate domain wall, are solvable as commuting projector Hamiltonian.

## 3.2 Fermionic anyon condensation

### 3.2.1 Fermionic anyon condensation in UMTC $\mathcal{B}$

A topological order can encompass a fermion $f$ as an object characterized by nontrivial self-braiding. In a braided tensor category $\mathcal{B}$, it's possible to carry out fermionic anyon condensation following the methodology detailed in Ref. [92].

To perform fermion condensation in a UMTC $\mathcal{B}$, we introduce another minimal fermionic system $\mathcal{F}_0$ with two objects $1, \psi$, where $\psi$ is a simple fermion with fusion rule $\psi \times \psi = 1$. We then consider the stacked model $\mathcal{B} \boxtimes \mathcal{F}_0$ of $\mathcal{B}$ and $\mathcal{F}_0$, where a new boson $(f, \psi)$ arises from the fusion of the fermion $f$ in $\mathcal{B}$ and $\psi$ in $\mathcal{F}_0$. The fermionic anyon condensation is defined as the bosonic anyon condensation introduced previously by modding out the boson $(f, \psi)$ in $\mathcal{B} \boxtimes \mathcal{F}_0$. In other words, we choose the condensable algebra to be $A = (1, 1) \oplus (f, \psi)$ in $\mathcal{B} \boxtimes \mathcal{F}_0$, and

the condensed phase is given by the category $(\mathcal{B} \boxtimes \mathcal{F}_0)_A^0$. This condensed phase is not modular because the fermion $\psi$ is deconfined in the condensation process and has trivial full braidings with all anyons. The total quantum dimension of the condensed phase is related to that of the original phase by

$$\dim((\mathcal{B} \boxtimes \mathcal{F}_0)_A^0) = \frac{\dim \mathcal{B} \times \dim \mathcal{F}_0}{(\dim A)^2} = \frac{1}{2} \dim \mathcal{B}, \tag{64}$$

where $\dim \mathcal{F}_0 = 2$ and $\dim A = 2$.

### 3.2.2 Fermionic anyon condensation in UFC $\mathcal{C}$

Let's now explore the process of condensing a fermion within a fusion category $\mathcal{C}$. In this context, a fermion $f$ in $\mathcal{C}$ means that when lift to the level of the Drinfeld center $Z(\mathcal{C})$, it exhibits fermionic behavior.

The challenge arises due to the nontrivial self-braiding of fermions, preventing their arbitrary creation or annihilation within a vacuum. However, we can circumvent this issue by employing fermion creation and annihilation operators to terminate fermionic worldlines. This process elevates a fermionic object to the status of a fermionic morphism. Consequently, we can perform fermion condensation [57,60] within a fusion category, resulting in a super-fusion category — a category featuring $\mathbb{Z}_2$-graded Hom spaces.

If we consider the fusion category as the input for a Levin-Wen string-net model, the condensed super-fusion category becomes the input data in constructing a 3D fermionic string-net model. Hence, fermion condensation serves as a method to generate fermionic topological order from a bosonic one. The study of fermionic string-net models originated with the fermionic toric code model [67] and its extensions [68]. This concept evolved further to encompass Majorana-type fermionic string-net models [57,60]. It's worth noting that fermionic string-net models can also be interpreted as gauging of the bosonic subgroup of a fermionic symmetry-protected topological phase (FSPT). The classification of FSPT is addressed using group supercohomology theory [93–96] or spin cobordism [64], revealing both complex fermion and Majorana fermion layers within the FSPT classification.

In a super-fusion category, the Hom spaces exhibit a $\mathbb{Z}_2$-graded structure. The even and odd components correspond to bosonic and fermionic operators, respectively. Specifically, for an object $X$ within the super-fusion category, the $\mathrm{End}(X)$ space becomes a super division algebra, with only two distinct irreducible types: the complex numbers themselves $\mathbb{C}$ and the Clifford algebra $\mathbb{C}l_1 = \mathbb{C}^{1|1}$. Objects possessing $\mathrm{End}(X) = \mathbb{C}$ and $\mathbb{C}l_1$ are referred to as $m$-type and $q$-type objects, respectively [57].

Now, let's assume that there exists a fermion $f$ within a fusion category $\mathcal{C}$. This fermion has the fusion rule $f \times f = 1$. The algebra we intend to condense is $A^f = 1 \oplus f$. This algebra should be associative, meaning that the $\mathbb{Z}_2$ symmetry generated by $f$ is non-anomalous. Upon completion of the condensation process, a super-fusion category denoted as $\mathcal{C}_{A^f}$ emerges.

At the object level, the fermion $f$ is effectively identified with the vacuum object 1. In general, an object $X$ should be identified with $X \otimes f$ in the super-fusion category $\mathcal{C}_{A^f}$. So an object $[X]$ within $\mathcal{C}_{A^f}$ corresponds to $X \otimes A^f$ in terms of the original fusion category $\mathcal{C}$.

At the level of morphisms, analogous to the bosonic anyon condensation given by Eq. (57), we should have the following relation for fermionic anyon condensation:

$$\begin{aligned}
\mathrm{Hom}_{\mathcal{C}_{A^f}}(X \otimes A^f, Y \otimes A^f) &= \mathrm{Hom}_{\mathcal{C}}(X, Y \otimes A^f) \\
&= \mathrm{Hom}_{\mathcal{C}}(X, Y) \oplus \mathrm{Hom}_{\mathcal{C}}(X, Y \otimes f).
\end{aligned} \tag{65}$$

This equation connects the Hom spaces of the condensed super-fusion category to those of the original fusion category. From this relation, we can discern two types of objects that arise

after the condensation. The first type encompasses objects $a \in \mathcal{C}$ such that $a \not\simeq a \times f$. In this scenario, the endomorphism $\mathrm{End}_{\mathcal{C}_{Af}}([X]) = \mathrm{Hom}_{\mathcal{C}_{Af}}(X \otimes A^f, X \otimes A^f)$ is simply $\mathbb{C}$. This corresponds to the $m$-type object within $\mathcal{C}_{Af}$. Conversely, if $X$ remains unchanged under the fusion with $f$, then both terms on the right-hand side of Eq. (65) contribute. In this case, the endomorphism $\mathrm{End}_{\mathcal{C}_{Af}}([X])$ corresponds to $\mathbb{C}l_1 = \mathbb{C}^{1|1}$. Thus, the fixed point object $X$ with $X \simeq X \otimes f$ will lead to the condensation of a $q$-type object within $\mathcal{C}_{Af}$.

At the next level of natural transformations involving the associator, we can proceed to construct the $F$ symbols of the super-fusion category using the $F$ symbols of the original fusion category $\mathcal{C}$. A comprehensive explanation of this construction is provided in section 3.5 in the general $\mathbb{Z}_N$ case.

As an example, we can consider $\mathcal{C}$ to be the Ising category with objects $1, \psi$ and $\sigma$. The endomorphism of $[\sigma] = \sigma \otimes A^f$ after the condensation of $A^f = 1 \oplus \psi$ is

$$\mathrm{End}_{\mathcal{C}_{Af}}([\sigma]) = \mathrm{Hom}_{\mathcal{C}_{Af}}(\sigma \otimes A^f, \sigma \otimes A^f) = \mathrm{Hom}_{\mathcal{C}}(\sigma, \sigma \otimes A^f) = \mathrm{Hom}_{\mathcal{C}}(\sigma, \sigma \otimes (1 \oplus \psi))$$

$$= \mathrm{Hom}_{\mathcal{C}}(\sigma, \sigma \oplus (\sigma \otimes \psi)) = \mathrm{Hom}_{\mathcal{C}}(\sigma, \sigma) \oplus \mathrm{Hom}_{\mathcal{C}}(\sigma, \sigma \otimes \psi) = \mathbb{C}^{1|1}. \tag{66}$$

So $[\sigma] = \sigma \otimes A \in \mathcal{C}_{Af}$ is a $q$-type object. The condensed super-fusion category has only two simple objects $[1]$ and $[\sigma]$.

The total quantum dimension of the super-fusion category can be defined as

$$\dim(\mathcal{C}_{Af}) = \sum_{a \in \mathcal{C}_{Af}} \frac{d_a^2}{\dim \mathrm{End}(a)} = \sum_{a \in m \text{ type}} d_a^2 + \frac{1}{2} \sum_{a \in q \text{ type}} d_a^2. \tag{67}$$

By segregating the objects in $\mathcal{C}$ into those that are either non-fixed or fixed points of fusion $f$, it becomes evident that the dimensions prior to and post the fermion condensation are related by

$$\dim(\mathcal{C}_{Af}) = \frac{1}{2} \dim(\mathcal{C}). \tag{68}$$

It exhibits similarities to the dimension of fermion condensation in a UMTC $\mathcal{B}$, as seen in Eq. (64).

We can also explore fermion condensation in a fusion category from the perspective of 3d TQFT. Similar to previous discussions, the original fusion category $\mathcal{C}$ serves as the input for the Hamiltonian of the Levin-Wen string-net model. After fermion condensation, the resulting super-fusion category $\mathcal{C}_{Af}$ can be employed to construct a fermionic string-net model [60]. The excitations in this model should be described by a fermionic analog of the Drinfeld center, denoted as $Z_f(\mathcal{C}_{Af})$. In practice, tube algebras can be used as tools to analyze the properties of these fermionic excitations [57, 67].

The fermionic condensation from the fusion category $\mathcal{C}$ to the super-fusion category $\mathcal{C}_{Af}$ implies the existence of a gapped domain wall between the bosonic Levin-Wen string-net model $Z(\mathcal{C})$ and the fermionic string-net model $Z_f(\mathcal{C}_{Af})$. This raises the question of whether fermion condensation in the fusion category $\mathcal{C}$ can be connected to fermion condensation in the bulk UMTC $Z(\mathcal{C})$. In the special case where $\mathcal{C} = \mathcal{B}$ is a UMTC with a fermion, we have a relation [57]:

$$Z_f(\mathcal{B}_{Af}) = \mathcal{B} \boxtimes (\mathcal{B}_{Af}), \tag{69}$$

However, in the general scenario of a fusion category, this relation remains to be established.

## 3.3   Parafermionic anyon condensation in UMTC $\mathcal{B}$

In a manner similar to fermion condensation in a braided category [92], we can extend the concept of condensation from $\mathbb{Z}_2$ fermions to $\mathbb{Z}_N$ parafermions.

Let us assume that $\mathcal{B}$ is a UMTC. There is a $\mathbb{Z}_N$ parafermion labeled as $f$ with the fusion rule $f^N = 1$. So there is a subcategory of $\mathcal{B}$ with fusion ring $\mathbb{Z}_N$. As elaborated in Appendix B, a comprehensive classification of braided tensor categories with a fusion ring corresponding to $\mathbb{Z}_N$ can be derived. This classification consists of pre-metric groups $\mathcal{C}(\mathbb{Z}_N, k)$ associated with group $\mathbb{Z}_N$, where $k$ is an integer falling within a certain range that depends on the parity of $N$.

When $N$ is an odd integer, the parameter $k$ of the pre-metric group $\mathcal{C}(\mathbb{Z}_N, k)$ can range from 0 to $N-1$. This results in a set of $N$ distinct braided tensor categories, each characterized by a fusion ring following the $\mathbb{Z}_N$ structure. In contrast, when $N$ is an even integer, $k$ has a broader range of values, spanning from 0 to $2N - 1$.

As detailed in Appendix B, once a value for $k$ is specified, all the fusion and braiding properties become determined and well-defined. The modular matrices that govern these properties are also established. In particular, the $F$ symbol in Eq. (166) serves as a tool to comprehend the anomaly conditions associated with the $\mathbb{Z}_N$ symmetry:

$$\text{non-anomalous subcategory } \mathcal{C}(\mathbb{Z}_N, k) \Longleftrightarrow N \text{ is odd or } k \text{ is even.} \tag{70}$$

Hence, determining the precise value of the parameter $k$ within a given UMTC $\mathcal{B}$ holds significant significance. The simplest method to accomplish this is by examining the topological spin of the $\mathbb{Z}_N$ generator. Its relationship with $k$ is expressed as follows:

$$h_1 = \frac{k}{\gcd(2, N)N}, \tag{71}$$

as shown in Eq (165).

Now, let's assume that the subcategory $\mathcal{C}(\mathbb{Z}_N, k)$ within $\mathcal{B}$ is non-anomalous. Similar to the process of fermion condensation described in [92], we can introduce another parafermionic system denoted as $\mathcal{F}_0 = \mathcal{C}(\mathbb{Z}_N, -k)$. This system consists of $N$ objects $1, \psi, ..., \psi^{N-1}$. Notably, the generator $\psi$ represents a parafermion with a fusion rule of $\psi^N = 1$, and it possesses an opposite spin compared to the previous parafermion $f$. By adopting this approach, the composite model $\mathcal{B} \boxtimes \mathcal{F}_0$ exhibits a boson $(f, \psi)$.

The concept of parafermionic anyon condensation is established by employing the same principles as bosonic anyon condensation, but applied to the boson $(f, \psi)$ within $\mathcal{B} \boxtimes \mathcal{F}_0$. This entails condensing the algebra $A = (1, 1) \oplus (f, \psi) \oplus ... \oplus (f^{N-1}, \psi^{N-1})$ within $\mathcal{B} \boxtimes \mathcal{F}_0$, resulting in the emergence of a condensed phase characterized by the category $(\mathcal{B} \boxtimes \mathcal{F}_0)_A^0$. By a straightforward calculation, the total quantum dimension of the condensed phase is given by the formula:

$$\dim((\mathcal{B} \boxtimes \mathcal{F}_0)_A^0) = \frac{\dim \mathcal{B} \times \dim \mathcal{F}_0}{(\dim A)^2} = \frac{1}{N} \dim \mathcal{B}, \tag{72}$$

where $\dim \mathcal{F}_0 = N$ and $\dim A = N$. It generalizes the fermion case in a straightforward manner.

A special case arises when we have a specific value of $k$ such that $\mathcal{C}(\mathbb{Z}_N, k)$ becomes a modular category. As shown in Appendix A, this condition can be stated as follows:

$$\mathcal{C}(\mathbb{Z}_N, k) \text{ is modular} \Longleftrightarrow q_k \text{ is non-degenerate} \Longleftrightarrow \gcd(k, N) = 1. \tag{73}$$

If the UMTC $\mathcal{B}$ contains such $\mathbb{Z}_N$ subcategory that is modular, then $\mathcal{B}$ can be decomposed as follows:

$$\mathcal{B} = \mathcal{C}(\mathbb{Z}_N, k) \boxtimes \mathcal{B}', \tag{74}$$

where $\mathcal{B}'$ is another subcategory of $\mathcal{B}$ that is also modular. Since the anyons in $\mathcal{C}(\mathbb{Z}_N, k)$ and $\mathcal{B}'$ are unrelated and have trivial mutual braidings, the parafermion condensation is independent of $\mathcal{B}'$. Therefore, the condensed phase is solely determined by the parafermion condensation within the category $\mathcal{C}(\mathbb{Z}_N, k)$:

$$(\mathcal{B} \boxtimes \mathcal{F}_0)_A^0 = (\mathcal{C}(\mathbb{Z}_N, k) \boxtimes \mathcal{B}' \boxtimes \mathcal{F}_0)_A^0 = (\mathcal{C}(\mathbb{Z}_N, k) \boxtimes \mathcal{F}_0)_A^0 \boxtimes \mathcal{B}' = \mathcal{B}'. \tag{75}$$

In the last step, we start by identifying $f^i$ with $\psi^{N-i}$ for $1 \leq i \leq N-1$ through the condensation of $A$. This implies that the right $A$-module $(\mathcal{C}(\mathbb{Z}_N, k) \boxtimes \mathcal{F}_0)_A$ becomes equivalent to $\mathcal{C}(\mathbb{Z}_N, k)$. As this category is modular, all nontrivial objects possess nontrivial full braidings with $f$. Consequently, the only deconfined anyon in this context is the vacuum. So $(\mathcal{C}(\mathbb{Z}_N, k) \boxtimes \mathcal{F}_0)_A^0$ is trivial. In summary, when the $\mathbb{Z}_N$ part of $\mathcal{B}$ is modular, the condensed phase simplifies to $\mathcal{B}'$ which can be considered orthogonal to $\mathbb{Z}_N$ due to the decomposition $\mathcal{B} = \mathcal{C}(\mathbb{Z}_N, k) \boxtimes \mathcal{B}'$.

### 3.4 Parafermionic anyon condensation in UFC $\mathcal{C}$

In this section, we will explore the parafermionic anyon condensation in a fusion category $\mathcal{C}$. Similar to the fermion condensation, we will also use the term "parafermion" to refer to an object $f$ that can be lifted to a parafermion within the Drinfeld center $Z(\mathcal{C})$.

As discussed in the previous section, the $\mathbb{Z}_N$ parafermion $f$ should generate an algebra $A^{pf} \simeq \mathcal{C}(\mathbb{Z}_N, k)$ which forms a braided tensor subcategory of $Z(\mathcal{C})$. Although the fusion category $\mathcal{C}$ might not exhibit braiding, the parafermion $f$ still has a half-braiding with all objects in $\mathcal{C}$.

The parafermion condensation can be understood as elevating the parafermionic object to the parafermionic ($\mathbb{Z}_N$-graded) Hom space. Consequently, after the condensation of $A^{pf}$, a fusion category $\mathcal{C}$ transforms into a para-fusion category denoted as $\mathcal{C}_{A^{pf}}$.

Before delving into the concept of condensation, it's important to establish an understanding of the simple objects within a para-fusion category. In a super-fusion category, we encountered both $m$-type and $q$-type simple objects, each associated with a distinct super division algebra for their endomorphisms. For para-fusion categories, the simple objects should possess $\mathbb{Z}_N$-graded division algebras as their endomorphism algebras. As outlined in Appendix B, these objects can be classified using $\mathbb{C}[\mathbb{Z}_M]$, where $M$ is a divisor of $N$. This classification introduces various layers of $q$-type objects within the para-fusion category.

Now, let's assume that there exists a $\mathbb{Z}_N$ parafermion $f$ within a fusion category $\mathcal{C}$. The algebra we intend to condense is $A^{pf} = \mathcal{C}(\mathbb{Z}_N, k) = 1 \oplus f \oplus ... \oplus f^{N-1}$. This algebra should be associative, meaning that the $\mathbb{Z}_N$ symmetry generated by $f$ is non-anomalous. This also give us the condition Eq. (70).

At the object level, the parafermion $f$ should be effectively treated as the vacuum object 1. In this context, any object $X$ gets identified with $X \otimes f$ within the para-fusion category $\mathcal{C}_{A^{pf}}$. This means that an object $[X]$ within $\mathcal{C}_{A^{pf}}$ essentially corresponds to $X \otimes A^{pf}$ in the original fusion category $\mathcal{C}$.

At the level of morphisms, similar to the bosonic anyon condensation discussed in Eq. (57), the relation for parafermionic anyon condensation can be expressed as follows:

$$\begin{aligned}
&\mathrm{Hom}_{\mathcal{C}_{A^{pf}}}(X \otimes A^{pf}, Y \otimes A^{pf}) \\
&= \mathrm{Hom}_{\mathcal{C}}(X, Y \otimes A^{pf}) \\
&= \mathrm{Hom}_{\mathcal{C}}(X, Y) \oplus \mathrm{Hom}_{\mathcal{C}}(X, Y \otimes f) \oplus ... \oplus \mathrm{Hom}_{\mathcal{C}}(X, Y \otimes f^{N-1}).
\end{aligned} \tag{76}$$

This equation establishes a connection between the Hom spaces of the condensed para-fusion category and those of the original fusion category.

Moving to the next step, we delve into the realm of natural transformations that involve the associator. In this phase, we embark on constructing the $F$ symbols of the para-fusion category using the pre-existing $F$ symbols derived from the original fusion category $\mathcal{C}$. To fully comprehend this intricate process, we consolidate the entire construction in section 3.5.

Similar to the super-fusion category, we can define the total quantum dimension of a para-fusion category as follows:

$$\dim(\mathcal{C}_{A^{pf}}) = \sum_{a \in \mathcal{C}_{A^{pf}}} \frac{d_a^2}{\dim \mathrm{End}(a)}. \tag{77}$$

The connection between the dimensions before and after parafermion condensation can be expressed as:

$$\dim(\mathcal{C}_{A^{pf}}) = \frac{\dim(\mathcal{C})}{N}. \tag{78}$$

From a 3d TQFT perspective, we can represent the Levin-Wen string-net model on the left using the fusion category $\mathcal{C}$ as input data. On the right-hand side, the para-fusion category $\mathcal{C}_{A^{pf}}$ corresponds to a parafermionic string-net model. Between these two models, there exists a gapped domain wall. This domain wall serves as a boundary separating the two phases, so it is a $(\mathcal{C}, \mathcal{C}_{A^{pf}})$-bimodule category. However, the specific relationship between anyon condensation in the fusion category $\mathcal{C}$ and the analogous process in $Z(\mathcal{C})$ still requires further investigation and study.

### 3.5 A functor from fusion category to para-fusion category

Now using the parafermionic condensation, we can establish a relationship between the $F$-symbols of para-fusion category and its parent fusion one. More specifically, consider the following move:

$$\tag{79}$$

where lines of $[\mathcal{I}]$ to $[\mathcal{N}]$ are TDLs in the para-fusion category condensed from lines of $\mathcal{I}$ to $\mathcal{N}$ in the parent fusion category, and the parafermionic defects $a, b, c$ and $d$ are the reminiscent of the condensed $\mathbb{Z}_N$-anyons in the parent theory. We aim to find how $\mathcal{F}_{[\mathcal{L}]}^{[\mathcal{I}],[\mathcal{J}],[\mathcal{K}]}$ can be obtained from the data of its parent fusion category. The stradegy is to lift the LHS of (79) back to the original category, then perform a series of F-moves, and finally condense back to

the para-fusion one in the RHS of (79):

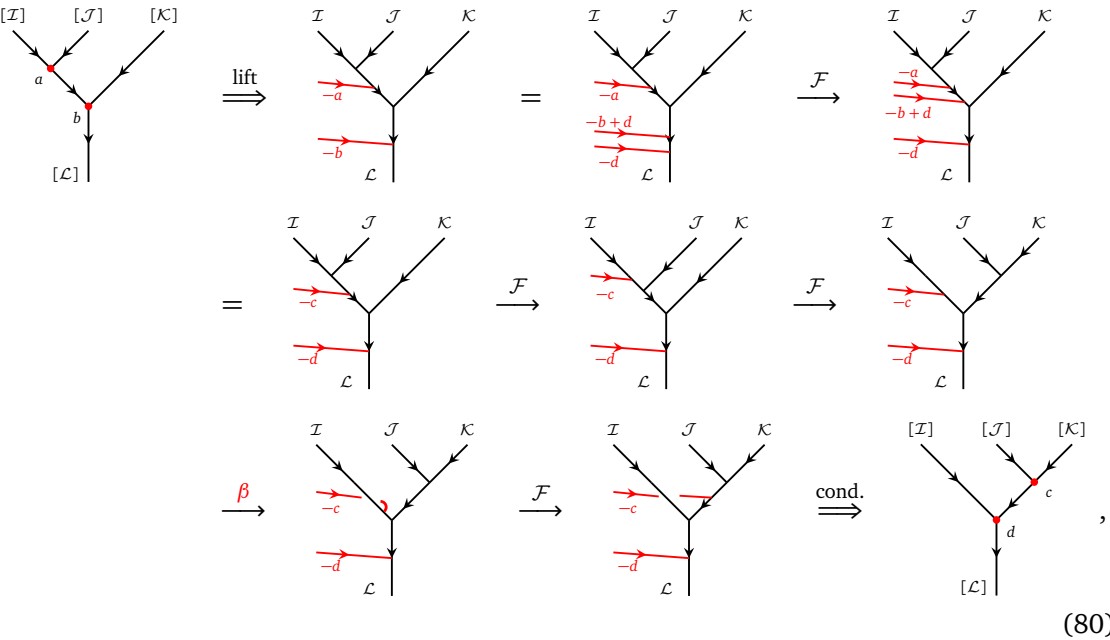

(80)

where the braiding operation "$\beta$" in the third line requires the knowledge of half braidings between the to-be-condensed $\mathbb{Z}_N$-anyons and the others defined through a morphism

$$\beta \in \mathrm{Hom}\left(a \otimes \mathcal{L}, \mathcal{L} \otimes a\right),\tag{81}$$

i.e.

$$\equiv \beta_a(b) \qquad \text{and} \qquad \equiv \beta_a(\mathcal{L}).\tag{82}$$

The half braiding data are encoded in the Drinfeld center $Z(\mathcal{C})$ of the parent fusion category $\mathcal{C}$. Therefore, to sum up, using (80) we establish a map from the $F$-symbols and half-braiding data of the fusion category $\mathcal{C}$ to the $F$-symbols in the para-condensed fusion category $\mathcal{C}/\mathbb{Z}_N$. In practice, we can either apply (80) or directly solve para-pentagon equations (54) to find the $F$-moves in the para-fusion category. It would be interesting to show the equivalence between (54) and (80) that we will leave in the future work. On the other hand, in the following sections, we use both of them to study various examples and demonstrate the consistency of the two approaches case by case.

## 4 Examples: Verlinde Lines

### 4.1 Parafermionic $\mathbb{Z}_3$

The first example for us to demonstrate the para-fusion category is the Verlinde lines in the parafermionization of 3-Potts model. As discussed at the end of section **??**, the 3-Potts model has partition function

$$Z_\mathcal{T} = |\chi_0|^2 + |\chi_{2/5}|^2 + |\chi_{1/15}|^2 + |\chi_{1/15^*}|^2 + |\chi_{2/3}|^2 + |\chi_{2/3^*}|^2.\tag{83}$$

Correspondingly, one has six Verlinde lines, denoted by

$$\mathcal{C} = \left\{\mathcal{L}_0, \mathcal{L}_{2/3}, \mathcal{L}_{2/3^*}, \mathcal{L}_{2/5}, \mathcal{L}_{1/15}, \mathcal{L}_{1/15^*}\right\},\tag{84}$$

where the first three lines form the $\mathbb{Z}_3$ symmetry, while the rest are non-invertible lines. The fusion rules in $\mathcal{C}$ can be easily obtained by Verlinde formula, from which one can study the orbits of $\{\mathcal{L}_{2/5}, \mathcal{L}_{1/15}, \mathcal{L}_{1/15^*}\}$ under the $\mathbb{Z}_3$-action. It turns out that

$$\mathcal{L}_{2/3} \cdot \mathcal{L}_{2/5} = \mathcal{L}_{1/15}, \quad \mathcal{L}_{2/3} \cdot \mathcal{L}_{1/15} = \mathcal{L}_{1/15^*}, \quad \mathcal{L}_{2/3} \cdot \mathcal{L}_{1/15^*} = \mathcal{L}_{2/5}, \tag{85}$$

implying that the set $\{\mathcal{L}_{2/5}, \mathcal{L}_{1/15}, \mathcal{L}_{1/15^*}\}$ form a 3-orbit, i.e. an orbit with 3 steps, under the $\mathbb{Z}_3$-action. Therefore, after parafermionization, the three Verlinde lines will condense to a single line in the $\mathbb{Z}_3$-parafermionic theory, whose fusion rule can be once again determined by its parent 3-Potts theory. Notice that

$$\mathcal{L}_{2/5} \cdot \mathcal{L}_{2/5} = \mathcal{L}_0 + \mathcal{L}_{2/5}. \tag{86}$$

The sub-category

$$\mathcal{C}' = \{\mathcal{L}_0, \mathcal{L}_{2/5}\} \tag{87}$$

is of Fibonacci type. If we choose $\mathcal{L}_0$ and $\mathcal{L}_{2/5}$ as the representatives after condensing the $\mathbb{Z}_3$ lines, the para-fusion category

$$\widetilde{\mathcal{C}}_1 \equiv \{[\mathcal{L}_0], [\mathcal{L}_{2/5}]\} \tag{88}$$

is isomorphic to $\mathcal{C}'$ with the fusion rule

$$[\mathcal{L}_{2/5}] \cdot [\mathcal{L}_{2/5}] = \mathbb{C}^{1|0|0} [\mathcal{L}_0] + \mathbb{C}^{1|0|0} [\mathcal{L}_{2/5}], \tag{89}$$

where $\mathbb{C}^{1|0|0}$ implies that the TDL $[\mathcal{L}_{2/5}]$ is of m-type, and the trivial parafermionic defect $\psi_0$ is located at the junctions of trivalent graphs for the fusion of TDLs. Since there are only trivial parafermionic defects in the choice of the representatives, the $F$-symbols satisfy usual pentagon identities, and the only non-trivial $F$-symbol is $\mathcal{F}^{[\mathcal{L}_{2/5}],[\mathcal{L}_{2/5}],[\mathcal{L}_{2/5}]}_{[\mathcal{L}_{2/5}]}$ given by,

$$\tag{90}$$

where the dashed and solid lines denote the TDLs $[\mathcal{L}_0]$ and $[\mathcal{L}_{2/5}]$ respectively, $\zeta = \frac{\sqrt{5}+1}{2}$, and we have uses the gauge freedom to fix the entry of the $F$-symbol,

$$\mathcal{F}^{[\mathcal{L}_{2/5}],[\mathcal{L}_{2/5}],[\mathcal{L}_{2/5}]}_{[\mathcal{L}_{2/5}]}([\mathcal{L}_0], [\mathcal{L}_{2/5}]) = 1. \tag{91}$$

The $F$-symbol in (91) is related to the standard $F$-symbol of Fibonacci Anyons by a similarity transformation (see for example [97]).

Instead, one can choose a different representatives, say $[\mathcal{L}_0]$ and $[\mathcal{L}_{1/15}]$ for example, denoted as

$$\widetilde{\mathcal{C}}_2 \equiv \{[\mathcal{L}_0], [\mathcal{L}_{1/15}]\} \tag{92}$$

Notice the fusion rule

$$\mathcal{L}_{1/15} \cdot \mathcal{L}_{1/15} = \mathcal{L}_{1/15^*} + \mathcal{L}_{2/3^*} = \mathcal{L}_{2/3} \cdot \mathcal{L}_{1/15} + \mathcal{L}_{2/3^*} \cdot \mathcal{L}_0, \tag{93}$$

implying the following fusion rule after condensation

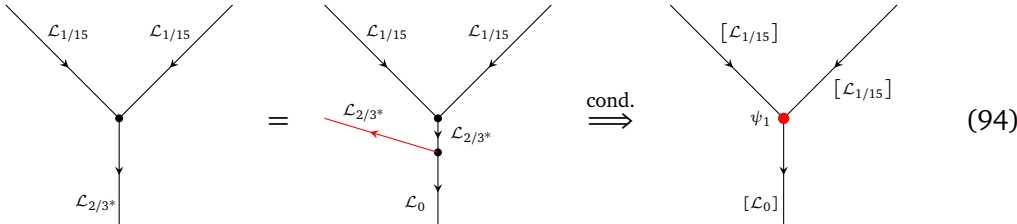

$$(94)$$

and

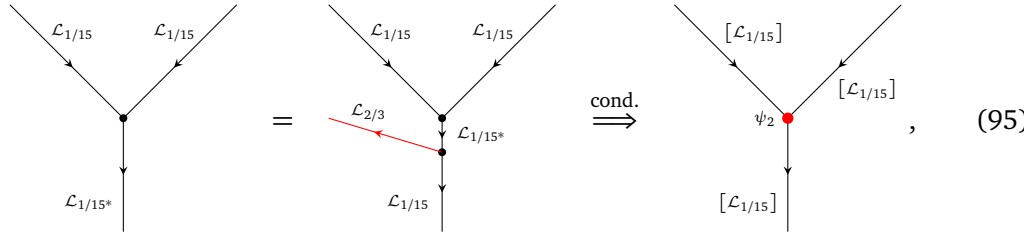

$$(95)$$

where the orientation of the red lines need to be further inverted to condense them in according to (85). The red dots $\psi_i$ denote the 1d parafermionic defects as the remnants of condensing the $\mathbb{Z}_3$-symmetry. They also specify the morphisms, $\mathrm{Hom}(\mathcal{L}_i \otimes \mathcal{L}_j, \mathcal{L}_k)$ of the fusion of two TDLs in the parafermoinic theory. In the case at hand, we have

$$\left[\mathcal{L}_{1/15}\right] \cdot \left[\mathcal{L}_{1/15}\right] = \mathbb{C}^{0|1|0} \left[\mathcal{L}_0\right] + \mathbb{C}^{0|0|1} \left[\mathcal{L}_{1/15}\right]. \tag{96}$$

With the above fusion rule, we can solve the para-pentagon equation with the choice of

$$\omega = e^{\frac{2\pi i}{3}}, \tag{97}$$

and find the solutions to the non-trivial $F$-symbols $\mathcal{F}$,

$$
\psi_2 \quad \psi_1 \quad = \quad \omega \quad \psi_2 \quad \psi_1, \tag{98}
$$

and

$$
\begin{pmatrix} \psi_1 & \psi_0 \\ \psi_2 & \psi_2 \end{pmatrix} = \begin{pmatrix} \omega^{-1}\zeta^{-1} & 1 \\ \zeta^{-1} & -\omega\zeta^{-1} \end{pmatrix} \cdot \begin{pmatrix} \psi_1 & \psi_0 \\ \psi_2 & \psi_2 \end{pmatrix}, \tag{99}
$$

where the dashed and solid lines stand for the TDLs $\left[\mathcal{L}_0\right]$ and $\left[\mathcal{L}_{1/15}\right]$ respectively. Solving the para-pentagons, one can find two solutions with $\zeta = \frac{1 \pm \sqrt{5}}{2}$. Spin selection rules [10] imply that $\zeta = \frac{1+\sqrt{5}}{2}$ and $\zeta = \frac{1-\sqrt{5}}{2}$ correspond to the existence of defect operators with spin $s = \pm\frac{2}{5}$ and $s = \pm\frac{1}{5}$ respectively. Therefore the solution with $\zeta = \frac{1+\sqrt{5}}{2}$ need to be picked up.

**Quantum dimensions.** The quantum dimensions of $[\mathcal{L}_{2/5}]$ and $[\mathcal{L}_{1/15}]$ can be determined through the largest eigenvalue of their fusion matrices, respectively. In the case of $\mathcal{L}_{2/5}$ this is straightforward and gives $d_{[2/5]} = \zeta$. In the case of $\mathcal{L}_{1/15}$ the fusion matrix is

$$N_{[\mathcal{L}_{1/15}]}{}^a{}_b = \begin{pmatrix} 0 & 1 \\ \omega & \omega^2 \end{pmatrix}, \quad a, b \in \{\mathcal{L}_0, \mathcal{L}_{1/15}\}, \tag{100}$$

and hence the eigenvalues are $\omega^{-1}\frac{1+\sqrt{5}}{2}$ and $\omega^{-1}\frac{1-\sqrt{5}}{2}$. The absolute value of the largest eigenvalue then gives

$$d_{\mathcal{L}_{1/15}} = \frac{1+\sqrt{5}}{2} = \zeta. \tag{101}$$

**Isomorphism of $\mathcal{C}_1$ and $\mathcal{C}_2$** As $\mathcal{C}_1$ and $\mathcal{C}_2$ are different only due to different choices of representatives, they are expected to be isomorphic to each other. One can establish a natural transformation between them by lifting them to the parent fusion category and then condense back. For example, for (98), we lift the LHS of the equation, perform a series of F-moves in the parent fusion category, and finally condense back to the RHS of (98):

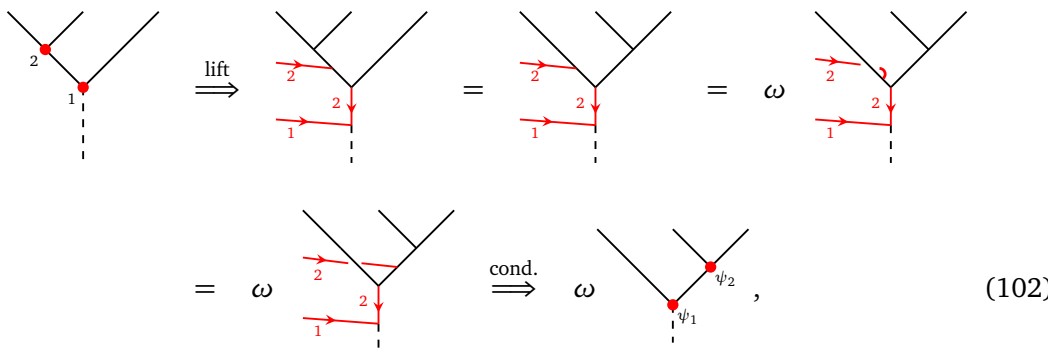

$$\tag{102}$$

where the prefactor "$\omega$" is precisely captured by the half-braiding between lines $\mathcal{L}_{\frac{1}{15}}$ and $\mathcal{L}_{\frac{2}{3}*}$. One can similarly work out the map from (89) to (99) that we will omit for brevity, and thus establish the isomorphism between $\mathcal{C}_1$ and $\mathcal{C}_2$.

## 4.2 Parafermionic $\mathbb{Z}_4$

Our second exmaple is the $\mathbb{Z}_4$ parafermionization of the $\mathbb{Z}_2$ orbifold theory of the $c = 1$ compact boson at special radius $R = \sqrt{6}$ which can be also realized as a $SU(2)_4/U(1)$ coset model. The original bosonic CFT is rational, and has 10 primaries with conformal weights,

$$\left\{0, \frac{3}{4}, 1, \frac{3}{4}^*, \frac{1}{16}, \frac{1}{16}^*, \frac{9}{16}, \frac{9}{16}^*, \frac{1}{3}, \frac{1}{12}\right\}, \tag{103}$$

where the "$*$" denotes charge conjugation as before. The 10 primaries implies that there are 10 Verlinde lines labeled by the conformal weights as well,

$$\mathcal{C} \equiv \left\{\mathcal{L}_0, \mathcal{L}_{\frac{3}{4}}, \mathcal{L}_1, \mathcal{L}_{\frac{3}{4}*}, \mathcal{L}_{\frac{1}{16}}, \mathcal{L}_{\frac{1}{16}*}, \mathcal{L}_{\frac{9}{16}}, \mathcal{L}_{\frac{9}{16}*}, \mathcal{L}_{\frac{1}{3}}, \mathcal{L}_{\frac{1}{12}}\right\}. \tag{104}$$

By studying their fusion rules, one can find the first four lines in $\mathcal{C}$ form the to-be-condensed $\mathbb{Z}_4$ symmetry. On the other hand, the rest are non-invertible TDLs, and can be grouped by the orbits of the $\mathbb{Z}_4$-actions. One finds that the set

$$\mathcal{S}_1 = \left\{\mathcal{L}_{\frac{1}{16}}, \mathcal{L}_{\frac{1}{16}*}, \mathcal{L}_{\frac{9}{16}}, \mathcal{L}_{\frac{9}{16}*}\right\} \tag{105}$$

form a 4-orbit, and the set

$$\mathcal{S}_2 = \left\{ \mathcal{L}_{\frac{1}{3}}, \mathcal{L}_{\frac{1}{12}} \right\} \tag{106}$$

form a 2-orbit, i.e. the two elements in $\mathcal{S}_2$ are $\mathbb{Z}_2$ fixed points out of $\mathbb{Z}_4$.

Now, after we parafermionize the bosonic theory to the $\mathbb{Z}_4$ parafermionic one, we end up with a para-fusion category consisting of three TDLs,

$$\widetilde{\mathcal{C}} = \left\{ [\mathcal{L}_0], \left[\mathcal{L}_{\frac{1}{3}}\right], \left[\mathcal{L}_{\frac{1}{16}}\right] \right\}, \tag{107}$$

where $[\mathcal{L}_0]$, $\left[\mathcal{L}_{\frac{1}{3}}\right]$ and $\left[\mathcal{L}_{\frac{1}{16}}\right]$ are representatives of the $Z_4$-symmetry category, lines in $\mathcal{S}_2$ and $\mathcal{S}_1$ respectively. Since $\mathcal{S}_2$ is a 2-orbit, the TDL $\left[\mathcal{L}_{\frac{1}{3}}\right]$ is a $\mathbb{Z}_2$ q-type object, whereas $\left[\mathcal{L}_{\frac{1}{16}}\right]$ is of m-type because elements in $\mathcal{S}_1$ are transitive with respect to $\mathbb{Z}_4$. The fusion rules of TDLs in $\widetilde{\mathcal{C}}$ can be read off from their parent category $\mathcal{C}$,

$$\begin{cases} \mathcal{L}_{\frac{1}{3}} \cdot \mathcal{L}_{\frac{1}{3}} = \mathcal{L}_0 + \mathcal{L}_1 + \mathcal{L}_{\frac{1}{3}}, \quad \mathcal{L}_{\frac{1}{3}} \cdot \mathcal{L}_1 = \mathcal{L}_{\frac{1}{3}} \\ \mathcal{L}_{\frac{1}{3}} \cdot \mathcal{L}_{\frac{1}{16}} = \mathcal{L}_{\frac{1}{16}} + \mathcal{L}_{\frac{9}{16}} = \mathcal{L}_{\frac{1}{16}} + \mathcal{L}_1 \cdot \mathcal{L}_{\frac{1}{16}}, \\ \mathcal{L}_{\frac{1}{16}} \cdot \mathcal{L}_{\frac{1}{16}} = \mathcal{L}_{\frac{1}{12}} + \mathcal{L}_{\frac{3}{4}^*} = \mathcal{L}_{\frac{3}{4}} \cdot \mathcal{L}_{\frac{1}{3}} + \mathcal{L}_{\frac{3}{4}^*} \cdot \mathcal{L}_0. \end{cases} \tag{108}$$

The first two equations in (108) simply imply that $\left[\mathcal{L}_{\frac{1}{3}}\right]$ is a $\mathbb{Z}_2$ q-type object after condensation. Put differently, the junctions between $\left[\mathcal{L}_{\frac{1}{3}}\right]$ and the other TDLs can have two types of parafermionic defect operators $\psi_0$ and $\psi_2$, i.e.

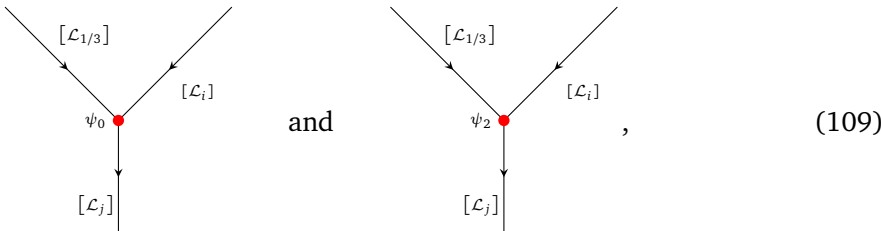

$$\tag{109}$$

where $[\mathcal{L}_i]$ and $[\mathcal{L}_j]$ are objects in $\widetilde{\mathcal{C}}$, and the presence of parafermionic defect operators $\psi_0$ and $\psi_2$ are due to the condensation of line $\mathcal{L}_0$ and $\mathcal{L}_1$. In the same fashion, the third equation in eq. (108) implies that

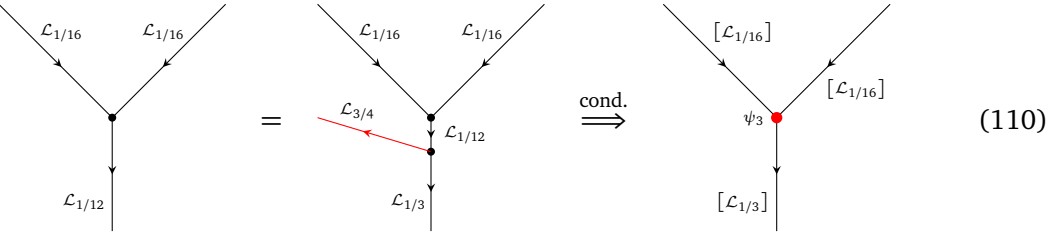

$$\tag{110}$$

and

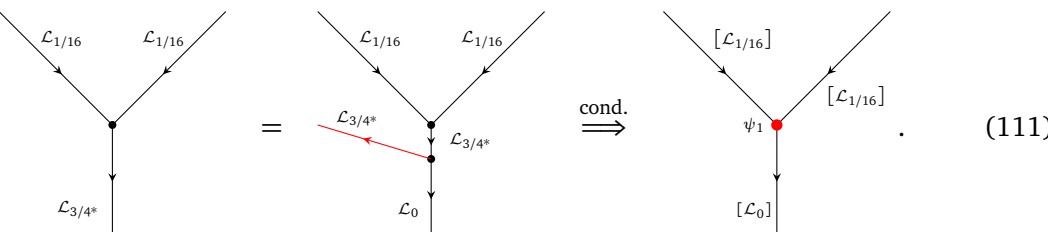

$$\tag{111}$$

Notice also in eq. (110), the TDL $\mathcal{L}_{1/3}$ is in addition invariant by fusing $\mathcal{L}_1$,

$$\mathcal{L}_{\frac{3}{4}} \cdot \mathcal{L}_{\frac{1}{3}} = \mathcal{L}_{\frac{3}{4}*} \cdot \mathcal{L}_{\frac{1}{3}} \tag{112}$$

Therefore after condensation, the parafermionic defect $\psi_1$ can be also located at the junction, i.e.

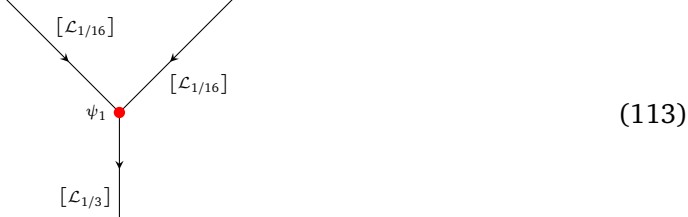

$$\tag{113}$$

To summarize, we have the following fusion rules for $\widetilde{\mathcal{C}}$:

$$
\begin{aligned}
&[\mathcal{L}_i] \cdot [\mathcal{L}_j] = [\mathcal{L}_j] \cdot [\mathcal{L}_i], \quad [\mathcal{L}_0] \cdot [\mathcal{L}_i] = [\mathcal{L}_i], \\
&[\mathcal{L}_{1/3}] \cdot [\mathcal{L}_{1/3}] = \mathbb{C}^{1|0|1|0}[\mathcal{L}_0] + \mathbb{C}^{1|0|1|0}[\mathcal{L}_{1/3}], \quad [\mathcal{L}_{1/3}] \cdot [\mathcal{L}_{1/16}] = \mathbb{C}^{1|0|1|0}[\mathcal{L}_{1/16}], \\
&[\mathcal{L}_{1/16}] \cdot [\mathcal{L}_{1/16}] = \mathbb{C}^{0|1|0|0}[\mathcal{L}_0] + \mathbb{C}^{0|1|0|1}[\mathcal{L}_{1/3}].
\end{aligned}
\tag{114}
$$

**Quantum dimensions.** Similarly to the $\mathbb{Z}_3$-case, we can solve for the quantum dimensions by studying the largest eigenvalue of the fusion matrix. Neglecting the dimensionality of the vertex, one finds from the above fusion rules

$$d_{[\mathcal{L}_0]} = 1, \quad d_{[\mathcal{L}_{1/3}]} = \frac{1+\sqrt{5}}{2}, \quad d_{[\mathcal{L}_{1/16}]} = \sqrt{2}. \tag{115}$$

Based on the fusion rule, we can solve the para-pentagon equations they need to satisfy. It turns out that there are 16 solutions to para-pentagons. One can compute their spin selection rules to further determine which one is for the parafermionic category $\widetilde{\mathcal{C}}$. Here, for brevity, we only spell out the two non-trivial F-moves with four legs of all $[\mathcal{L}_{1/3}]$ or $[\mathcal{L}_{1/16}]$. For $\mathcal{F}^{[\mathcal{L}_{1/3}],[\mathcal{L}_{1/3}],[\mathcal{L}_{1/3}]}_{[\mathcal{L}_{1/3}]}$, we found

$$
= \frac{1}{2}
\begin{pmatrix}
1 & 0 & 0 & -1 & 1 & 0 & 0 & -1 \\
0 & 1 & -1 & 0 & 0 & 1 & -1 & 0 \\
0 & 1 & -1 & 0 & 0 & -1 & 1 & 0 \\
1 & 0 & 0 & -1 & -1 & 0 & 0 & 1 \\
1 & 0 & 0 & 1 & 0 & 0 & 0 & 0 \\
0 & 1 & 1 & 0 & 0 & 0 & 0 & 0 \\
0 & -1 & -1 & 0 & 0 & 0 & 0 & 0 \\
-1 & 0 & 0 & -1 & 0 & 0 & 0 & 0
\end{pmatrix}
\cdot
\tag{116}
$$

and for $\mathcal{F}^{[\mathcal{L}_{1/16}],[\mathcal{L}_{1/16}],[\mathcal{L}_{1/16}]}_{[\mathcal{L}_{1/16}]}$ we have

$$
\begin{pmatrix} \vdots \end{pmatrix} \;=\; (-1)^{\frac{1}{4}} \begin{pmatrix} \frac{1}{\sqrt{3}} & \sqrt{\frac{2}{3}} & \sqrt{\frac{2}{3}} & 0 & 0 \\ \frac{1}{\sqrt{3}} & -\frac{1}{\sqrt{6}} & -\frac{1}{\sqrt{6}} & 0 & 0 \\ \frac{1}{\sqrt{3}} & -\frac{1}{\sqrt{6}} & -\frac{1}{\sqrt{6}} & 0 & 0 \\ 0 & 0 & 0 & \frac{i}{\sqrt{2}} & \frac{i}{\sqrt{2}} \\ 0 & 0 & 0 & \frac{i}{\sqrt{2}} & \frac{i}{\sqrt{2}} \end{pmatrix} \cdot \begin{pmatrix} \vdots \end{pmatrix}, \qquad (117)
$$

where the dotted, black and blue lines denote the $[\mathcal{L}_0]$, $[\mathcal{L}_{1/3}]$ and $[\mathcal{L}_{1/16}]$ TDLs respectively, and the red dots and numbers stand for the parafermionic defects $\psi_i$. One can verify that the ranks of $\mathcal{F}^{[\mathcal{L}_{1/3}],[\mathcal{L}_{1/3}],[\mathcal{L}_{1/3}]}_{[\mathcal{L}_{1/3}]}$ and $\mathcal{F}^{[\mathcal{L}_{1/16}],[\mathcal{L}_{1/16}],[\mathcal{L}_{1/16}]}_{[\mathcal{L}_{1/16}]}$ are 6 and 3. This is because the parafermionic defects can "move" along the $\mathbb{Z}_2$ q-type TDL $[\mathcal{L}_{1/3}]$.

## 4.3 Parafermionization of $SU(8)_4$

At last, we use the parafermionized $SU(8)_4$ to illustrate that its para-fusion category contains two $\mathbb{Z}_2$ q-type objects and one $\mathbb{Z}_4$ q-type object. Let us consider the Verlinde lines in $SU(8)_4$, which contains 330 simple objects characterized by the integrable representations of the affine Lie algebra $\widehat{\mathfrak{su}(8)}_4$. Although there are too many lines to spell out their $F$-symbols in details, one can still understand the structures of the para-fusion category after para-condensation of the $SU(8)_4$. First notice that, in these Verlinde lines, there are 8 simple currents corresponding to the non-anomalous $\mathbb{Z}_8$-symmetry. One can study all other TDLs under the action of the $\mathbb{Z}_8$-symmetry. The upshot is that the 330 TDLs are divided into three sectors consisted of forty 8-orbits, two 4-orbits, and a single 2-orbits. Therefore, after parafermionization respect to the $\mathbb{Z}_8$-symmetry, it suggests that we will have a para-fusion category containing forty $m$-type objects, two $\mathbb{Z}_2$ q-type objects, and a single $\mathbb{Z}_4$ q-type object.

# 5 Example: Para-condensed $TY(\mathbb{Z}_N)$ Category

Beside various Verlinde lines we have seen so far, there are much richer TDLs in these parafermionic $\mathbb{Z}_N$ theories. Among them, there is an interesting class of *non-Verlinde* TDLs, which only commute with the Virasora algebra and admit $N$ different types of parafermionic defects opeartor living on them. These TDLs can be obtained from para-condensation of the non-invertible duality defect $\mathcal{N}$-lines of their parent bosonic WZW model $\mathfrak{su}(2)_N/\mathfrak{u}(1)$. Recall that, at the conformal fixed points, these theories are self-dual with respect to gauging their $\mathbb{Z}_N$ symmetries. Therefore, one might gauge the $\mathbb{Z}_N$ symmetry only on the half plane [78], and the interface $\mathcal{N}$ between the gauged/ungauged planes turns out to be a non-invertible TDL, satisfying the fusion rule,

$$
\mathcal{N} \cdot \mathcal{N} = \sum_{a \in \mathbb{Z}_N} a \qquad \text{and} \quad \mathcal{N} \cdot a = a \cdot \mathcal{N} = \mathcal{N}, \quad \text{for} \quad a \in \mathbb{Z}_N. \qquad (118)
$$

The $\mathcal{N}$-line together with the $\mathbb{Z}_N$ symmetry combined are the well-known Tambara-Yamagami category $TY^{t,\kappa}_{\mathbb{Z}_N}$, where $t$, satisfying $1 \le t \le N-1$ and $\gcd(t, N) = 1$, defines a symmetric

bi-character $\chi_t$ as

$$\chi_t(a,b) = e^{\frac{2\pi i t a b}{N}} \equiv \omega_{N,t}^{ab}, \quad \text{for} \quad a,b \in \mathbb{Z}_N, \tag{119}$$

and $\kappa = \pm 1$ is the Frobius-Schur indicator. The $\mathbb{Z}_N$ TY category is completely determined by the two parameters $t$ and $\kappa$. The non-trivial $F$-symbols are given as below:

$$\tag{120}$$

where the black and red lines in the graphs denote the TDLs for $\mathcal{N}$-line and $\mathbb{Z}_N$-lines respectively. For the bosonic $\mathfrak{su}(2)_N/\mathfrak{u}(1)$ model, the $\mathcal{N}$-line combined with $\mathbb{Z}_N$ symmetry specify one of $\mathrm{TY}_{\mathbb{Z}_N}^{t,\kappa}$'s. For example, in the case of $\mathbb{Z}_2$, $\mathfrak{su}(2)_2/\mathfrak{u}(1) \simeq$ Ising, so the TY category is given by $\mathrm{TY}_{\mathbb{Z}_N}^{1,+1}$.

After para-condensation, the $\mathbb{Z}_N$-TDLs are all condensed to the trivial line. The category thus only contains two objects[2]

$$\{\mathbb{1}, \mathcal{N}\}, \tag{121}$$

with the fusion rule

$$\mathcal{N} \cdot \mathcal{N} = \mathbb{C}^{\overbrace{1|1|\cdots|1}^{N}} \mathbb{1}, \quad \text{and} \quad \mathcal{N} \cdot \mathbb{1} = \mathbb{1} \cdot \mathcal{N} = \mathcal{N} \tag{122}$$

Apparently, the fusion (122) condensed from (118) implies that there are $N$ different types of 1d parafermionic defects, denoted as "color", that can live on the $\mathcal{N}$-line.

In this subsection, we aim to propose a classification for all $\mathbb{Z}_N$ para-condensed $\mathrm{TY}_{\mathbb{Z}_N}^{t,\kappa}$, dubbed as "pf-$\mathrm{TY}_{\mathbb{Z}_N}^{t,\kappa,\beta}$", where the additional parameter $\beta$ will be explained below in length. Following the fusion rule, only those $F$-symbols with even number of $\mathcal{N}$-lines are non-vanishing,

$$\left\{ \mathcal{F}_{\mathbb{1}}^{\mathbb{1},\mathbb{1},\mathbb{1}}, \mathcal{F}_{\mathcal{N}}^{\mathbb{1},\mathbb{1},\mathcal{N}}, \mathcal{F}_{\mathcal{N}}^{\mathbb{1},\mathcal{N},\mathbb{1}}, \mathcal{F}_{\mathbb{1}}^{\mathbb{1},\mathcal{N},\mathcal{N}}, \mathcal{F}_{\mathcal{N}}^{\mathcal{N},\mathbb{1},\mathbb{1}}, \mathcal{F}_{\mathbb{1}}^{\mathcal{N},\mathbb{1},\mathcal{N}}, \mathcal{F}_{\mathbb{1}}^{\mathcal{N},\mathcal{N},\mathbb{1}}, \mathcal{F}_{\mathcal{N}}^{\mathcal{N},\mathcal{N},\mathcal{N}} \right\}. \tag{123}$$

Since, at trivalent junctions involving the $\mathcal{N}$-line, there could be $N$ different colors, these $F$-symbols are generically matrices. For example, for $\mathcal{F}_{\mathcal{N}}^{\mathbb{1},\mathbb{1},\mathcal{N}}$,

$$\tag{124}$$

---

[2]By abuse of notation $\mathcal{N}$ denotes the duality line in both the parent $\mathrm{TY}_{\mathbb{Z}_N}^{t,\kappa}$ category and its para-condensation.

where the dashed and solid lines denote the trivial and $\mathcal{N}$-line, respectively, and the red dot implies there are $N$ different colors at each junction. Therefore the $F$-symbol $\mathcal{F}_{\mathcal{N}}^{\mathbb{1};\mathbb{1},\mathcal{N}}$ is an $N \times N^2$ matrix. We denote the entries of the $F$-symbol as $\mathcal{F}_{\mathcal{L}}^{\mathcal{L},\mathcal{L},\mathcal{L}}(a,b,c,d)$. Notice that, because of the parafermionic parity (22),

$$\mathcal{F}_{\mathcal{L}}^{\mathcal{L},\mathcal{L},\mathcal{L}}(a,b,c,d) = 0, \qquad \text{if} \qquad a+b \neq c+d \mod N. \tag{125}$$

## 5.1 Half braiding structure

For parafermionic condensation of $\mathrm{TY}_{\mathbb{Z}_N}^{t,\kappa}$, we need to introduce a consistent braiding structure for some elements in $\mathrm{TY}_{\mathbb{Z}_N}^{t,\kappa}$. In fact, there is no such a structure for the whole $\mathrm{TY}_{\mathbb{Z}_N}^{t,\kappa}$ category except for $N = 2$. However a *half braiding* structure does exist for elements in $\mathbb{Z}_N$ and $\mathcal{N}$-line.

We define the braiding between $\mathbb{Z}_N$-anyons $a$ and $b$, and $\mathbb{Z}_N$-anyon $a$ and $\mathcal{N}$ as

where the braiding $\beta_a(b)$, for consistency, needs to be identical to the braiding structure of $\mathbb{Z}_N$-anyons, i.e.

$$\beta_a(b) = \omega_{N,t}^{ab} = e^{\frac{2\pi i t a b}{N}}. \tag{127}$$

On the other hand, for $\beta_a(\mathcal{N})$, first note that

where we have used the following F-move

in the third equality. Therefore we have, from (128),

$$\beta_a(\mathcal{N}) = \beta_1(\mathcal{N})^a \, \omega_{N,t}^{-\sum_{j=1}^{a-1} j} = \beta_1(\mathcal{N})^a \, \omega_{N,t}^{-\frac{a(a-1)}{2}}, \tag{130}$$

i.e. the braiding $\beta_1(\mathcal{N})$ determines all other braiding data between the $\mathcal{N}$- and $\mathbb{Z}_N$-anyons.

To find consistent braiding data $\beta_1(\mathcal{N})$, the naturality conditions have to be imposed:

The LHS of the above graph gives

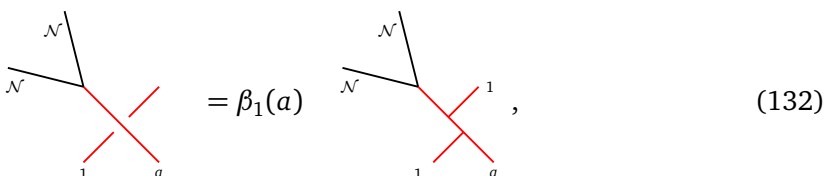

$$= \beta_1(a) \qquad , \tag{132}$$

while the RHS gives

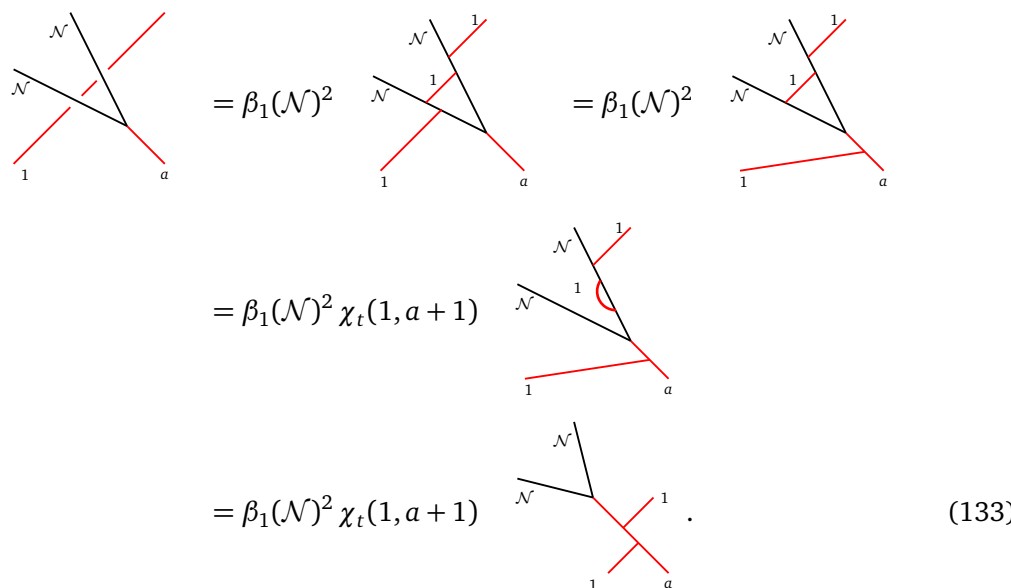

$$= \beta_1(\mathcal{N})^2 \qquad = \beta_1(\mathcal{N})^2$$

$$= \beta_1(\mathcal{N})^2 \chi_t(1, a+1)$$

$$= \beta_1(\mathcal{N})^2 \chi_t(1, a+1) \qquad . \tag{133}$$

Therefore, for consistency, we have

$$\beta_1(a) = \beta_1(\mathcal{N})^2 \chi_t(1, a+1) \quad \Longrightarrow \quad \beta_1(\mathcal{N}) = \pm e^{-\frac{\pi i t}{N}} , \tag{134}$$

as the allowable values, where we have used (127). Using (128), one arrives at

$$\beta_a(\mathcal{N}) = (\pm 1)^a \, \omega_{N,t}^{-\frac{a^2}{2}} . \tag{135}$$

We need to further require that

$$\beta_N(\mathcal{N}) = (\pm 1)^N (-1)^{tN} \equiv \beta_0(\mathcal{N}) = 1 . \tag{136}$$

Obviously, for even $N$, (136) will be automatically satisfied. Therefore $\beta_1(\mathcal{N})$ can be assigned $\pm e^{-\frac{\pi i t}{N}}$ as (134). On the other hand, for odd $N$, $\beta_1(\mathcal{N})$ has to be assigned as $(-1)^t e^{-\frac{\pi i t}{N}}$. In sum, we have

$$\beta \equiv \beta_1(\mathcal{N}) = \begin{cases} (-1)^t e^{-\frac{\pi i t}{N}} & \text{for} \quad \text{odd} \quad N \\ \pm e^{\frac{-\pi i t}{N}} & \text{for} \quad \text{even} \quad N \end{cases} \tag{137}$$

With these preparations, we can compute the braiding coefficient $B(a, \mathcal{N})$ introduced in (149). Notice that

$$= \quad = \beta_{-a}(\mathcal{N})$$

$$= \beta_{-a}(\mathcal{N}) \chi_t^{-1}(a, -a) \qquad = \beta_a(\mathcal{N}) \chi_t(a, a) \qquad . \tag{138}$$

We thus have

$$B(a, \mathcal{N}) = \beta_a(\mathcal{N}) \chi_t(a, a). \tag{139}$$

## 5.2 Parafermionic condensation

All of the $F$-symbols are constrained by the para-pentagon equations we proposed in sec.3. Subject to Ocneanu rigidity, after gauge fixings, there are only finite number of solutions. However, since the number of para-pentagon equations grows in terms of $\mathcal{O}(N^6)$, only for the case of $N = 2$, i.e. the fermionic case, can these para-pentagons be solved in such a brutal force way. Instead, we will use the data in $\mathrm{TY}_{\mathbb{Z}_N}^{t,\kappa}$ to simplify the $F$-symbols in its condensed pf-$\mathrm{TY}_{\mathbb{Z}_N}^{t,\kappa,\beta}$. During this procedure, a "half-braiding" structure has to be introduced as we will see soon.

We separate all $F$-symbols (123) into three sectors:

$$\begin{aligned}
\mathcal{S}_1 &= \left\{ \mathcal{F}_{\mathbb{1}}^{\mathbb{1},\mathbb{1},\mathbb{1}} \right\}, \\
\mathcal{S}_2 &= \left\{ \mathcal{F}_{\mathcal{N}}^{\mathbb{1},\mathbb{1},\mathcal{N}}, \mathcal{F}_{\mathcal{N}}^{\mathbb{1},\mathcal{N},\mathbb{1}}, \mathcal{F}_{\mathbb{1}}^{\mathbb{1},\mathcal{N},\mathcal{N}}, \mathcal{F}_{\mathcal{N}}^{\mathcal{N},\mathbb{1},\mathbb{1}}, \mathcal{F}_{\mathbb{1}}^{\mathcal{N},\mathbb{1},\mathcal{N}}, \mathcal{F}_{\mathbb{1}}^{\mathcal{N},\mathcal{N},\mathbb{1}} \right\}, \\
\mathcal{S}_3 &= \left\{ \mathcal{F}_{\mathcal{N}}^{\mathcal{N},\mathcal{N},\mathcal{N}} \right\}.
\end{aligned} \tag{140}$$

The $F$-symbol $\mathcal{F}_{\mathbb{1}}^{\mathbb{1},\mathbb{1},\mathbb{1}}$ in $\mathcal{S}_1$ is trivially solved by the para-pentagon consisted of five trivial lines,

$$\mathcal{F}_{\mathbb{1}}^{\mathbb{1},\mathbb{1},\mathbb{1}} = 1. \tag{141}$$

Now we will establish some linear relations among entries of each $F$-symbol in $\mathcal{S}_2$. Notice that the 1d parafermionic defects can "move" along the q-type $\mathcal{N}$-line. We can thus merge two different colors of defects by lifting them to the parent $\mathrm{TY}_{\mathbb{Z}_N}^{t,\kappa}$, performing F-moves, and then condensing back,

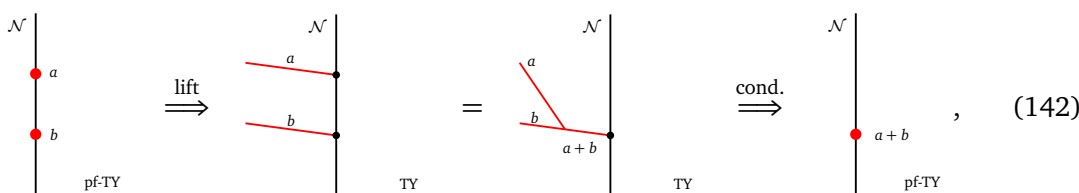

$$\tag{142}$$

where we have used the F-move (120) in $\mathrm{TY}_{\mathbb{Z}_N}^{t,\kappa}$.

For a trivalent junction in pf-$\mathrm{TY}_{\mathbb{Z}_N}^{t,\kappa,\beta}$, we can also use (142) to define its lifting in $\mathrm{TY}_{\mathbb{Z}_N}^{t,\kappa}$. Notice that there are three trivalent junctions involving $\mathcal{N}$-line from (122),

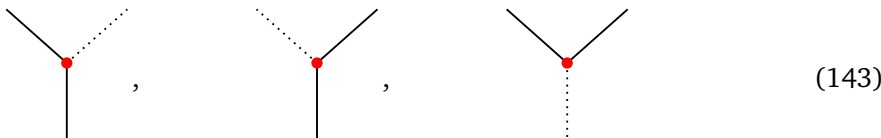

$$\tag{143}$$

There are ambiguities when one tries to lift the above junctions to $\mathrm{TY}_{\mathbb{Z}_N}^{t,\kappa}$. For example we can lift the first junction in (143) in two different ways as

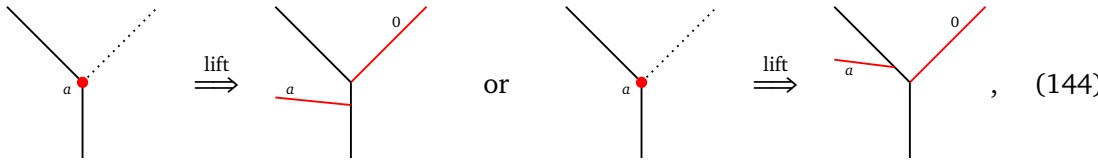

$$\tag{144}$$

where we have chosen $\mathbb{1} = [0]$ as the representative of $\mathbb{Z}_N$. However by the F-move in (120), we have

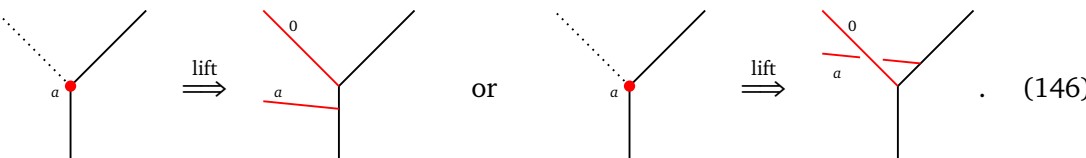

$$\tag{145}$$

Therefore there is actually no ambiguity for the lifting of (144). We now look at the second junction in (143), and have

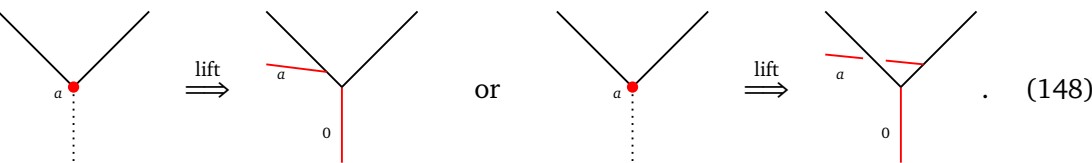

$$\tag{146}$$

The two ways of lifting can be related in the following F-move and braiding,

$$\tag{147}$$

where the coefficient $B(a,0)$ denotes the braiding of anyon "$a$" and trivial line, which is always unity. Therefore there is no ambiguity either in (146). Finally let's come to the third junction in (143), and its two ways of lifting,

$$\tag{148}$$

Similar to the previous calculation, we can establish the relation between the two liftings,

$$\tag{149}$$

where $B(a,\mathcal{N})$ is the braiding coefficient between anyon "$a$" and $\mathcal{N}$-line.

Because of the ambiguity in (148), let us assign the lifting of the trivalent junctions (143)

in the following way:

$$
\left\{
\begin{array}{c}
\text{(diagrams)}
\end{array}
\right.
\tag{150}
$$

Here we briefly remark that one can also use the second equation in (148) to define the lifting and condensation. The two ways are actually equivalent up to a gauge transformation by redefining the trivalent junction.

Now, using (150), we can proceed to establish various linear relations for the components of $F$-symbols in $\mathcal{S}_2$.

For $\mathcal{F}_{\mathcal{N}}^{\mathbb{1},\mathbb{1},\mathcal{N}}$,

$$
\implies \quad \mathcal{F}_{\mathcal{N}}^{\mathbb{1},\mathbb{1},\mathcal{N}}(0,a+b,a,b) = \mathcal{F}_{\mathcal{N}}^{\mathbb{1},\mathbb{1},\mathcal{N}}(0,a+b,0,a+b)
\tag{151}
$$

For $\mathcal{F}_{\mathcal{N}}^{\mathbb{1},\mathcal{N},\mathbb{1}}$,

and

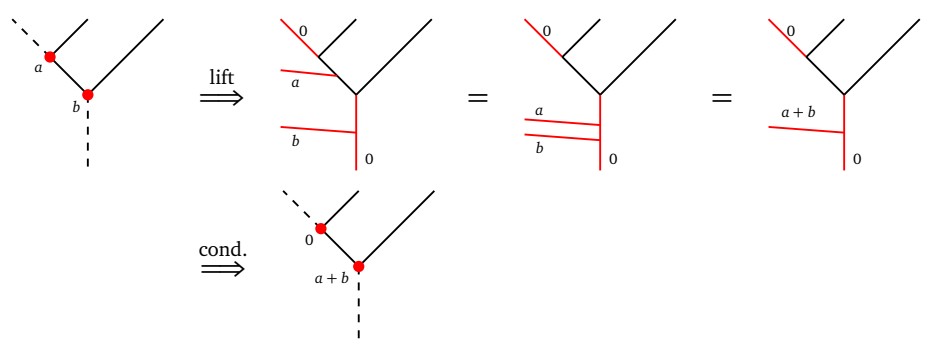

$$\implies \quad \mathcal{F}_{\mathcal{N}}^{\mathbb{1},\mathcal{N},\mathbb{1}}(a,b,c,d) = \mathcal{F}_{\mathcal{N}}^{\mathbb{1},\mathcal{N},\mathbb{1}}(0,a+b,0,c+d), \quad \text{for} \quad a+b = c+d \mod N \tag{152}$$

For $\mathcal{F}_{\mathbb{1}}^{\mathbb{1},\mathcal{N},\mathcal{N}}$,

$$\implies \quad \mathcal{F}_{\mathbb{1}}^{\mathbb{1},\mathcal{N},\mathcal{N}}(a,b,a+b,0) = \mathcal{F}_{\mathbb{1}}^{\mathbb{1},\mathcal{N},\mathcal{N}}(0,a+b,a+b,0) \tag{153}$$

For $\mathcal{F}_{\mathcal{N}}^{\mathcal{N},\mathbb{1},\mathbb{1}}$,

$$\implies \quad \mathcal{F}_{\mathcal{N}}^{\mathcal{N},\mathbb{1},\mathbb{1}}(a,b,0,a+b) = \mathcal{F}_{\mathcal{N}}^{\mathcal{N},\mathbb{1},\mathbb{1}}(0,a+b,0,a+b) \tag{154}$$

For $\mathcal{F}_{\mathbb{1}}^{\mathcal{N},\mathbb{1},\mathcal{N}}$,

$$(155)$$

and

$$\implies \quad \mathcal{F}_{\mathbb{1}}^{\mathcal{N},\mathbb{1},\mathcal{N}}(a,b,c,d) = B(c,\mathcal{N})\chi_t(c,d)F_{\mathbb{1}}^{\mathcal{N},\mathbb{1},\mathcal{N}}(0,a+b,0,c+d),$$
$$\text{for} \quad a+b = c+d \mod N \qquad (156)$$

At last, for $\mathcal{F}_{\mathbb{1}}^{\mathcal{N},\mathcal{N},\mathbb{1}}$,

$$\implies \quad \mathcal{F}_{\mathbb{1}}^{\mathcal{N},\mathcal{N},\mathbb{1}}(a+b,0,a,b) = B(a,\mathcal{N})\chi_t(a,b)\mathcal{F}_{\mathbb{1}}^{\mathcal{N},\mathbb{1},\mathcal{N}}(a+b,0,0,a+b) \qquad (157)$$

## 5.3   Gauge fixings and solutions to pf-$\mathrm{TY}^{t,\kappa,\beta}_{\mathbb{Z}_N}$

Using the condensation equations (152)−(158), one can dramatically simplify the para-pentagon equations. However there are many gauge freedoms to re-define various trivalent junctions. One has to fix all of them to fully solve all para-pentagons. For pf-$\mathrm{TY}^{t,\kappa,\beta}_{\mathbb{Z}_N}$, there are overall $3N-1$ gauge freedoms which can be counted by the method discussed in [61]. Among them, $3(N-1)$ gauges have been secretly fixed by (150), because choosing different condensations is equivalent to re-defining the trivalent junctions by using their corresponding gauge freedoms. The number $3(N-1)$ is due to the fact that there are three types of junctions with $N-1$ different colors except for the color "0" condensed from trivial line. Therefore specifying their condensations will uniquely fix $3(N-1)$ gauges. After that, we still have $3N-1-3(N-1)=2$ more gauges to fix. For convenience, we choose

$$\mathcal{F}^{\mathbb{1},\mathbb{1},\mathcal{N}}_{\mathcal{N}}(0,0,0,0) = \mathcal{F}^{\mathcal{N},\mathbb{1},\mathbb{1}}_{\mathcal{N}}(0,0,0,0) = \frac{1}{\sqrt{N}}\,. \tag{158}$$

Solving pentagons with two external $\mathcal{N}$-lines, one can find all F-moves in $\mathcal{S}_2$ as,

$$\begin{cases} \mathcal{F}^{\mathbb{1},\mathbb{1},\mathcal{N}}_{\mathcal{N}}(0,a+b,a,b) = \frac{1}{\sqrt{N}} \\[2mm] \mathcal{F}^{\mathbb{1},\mathcal{N},\mathbb{1}}_{\mathcal{N}}(a,b,c,d) = \frac{1}{N}\,, \quad \text{for} \quad a+b = c+d \mod N \\[2mm] \mathcal{F}^{\mathbb{1},\mathcal{N},\mathcal{N}}_{\mathbb{1}}(a,b,a+b,0) = \frac{1}{\sqrt{N}} \\[2mm] \mathcal{F}^{\mathcal{N},\mathbb{1},\mathbb{1}}_{\mathcal{N}}(a,b,0,a+b) = \frac{1}{\sqrt{N}} \\[2mm] \mathcal{F}^{\mathcal{N},\mathbb{1},\mathcal{N}}_{\mathbb{1}}(a,b,c,d) = \frac{1}{N}\,\beta_c(\mathcal{N})\,\chi_t(c,a+b), \quad \text{for} \quad a+b = c+d \mod N \\[2mm] \mathcal{F}^{\mathcal{N},\mathcal{N},\mathbb{1}}_{\mathbb{1}}(a,b,0,a+b) = \frac{1}{\sqrt{N}}\,\beta_a(\mathcal{N})\,\chi_t(a,a+b). \end{cases} \tag{159}$$

Finally, solving pentagons with four external $\mathcal{N}$-lines, we find the F-move $\mathcal{F}^{\mathcal{N},\mathcal{N},\mathcal{N}}_{\mathcal{N}}$ in $\mathcal{S}_4$,

$$\mathcal{F}^{\mathcal{N},\mathcal{N},\mathcal{N}}_{\mathcal{N}}(a,b,c,d) = \frac{\kappa}{\sqrt{N}}\,\beta_c(\mathcal{N})\,\chi_t(a,c), \quad \text{for} \quad a+b = c+d \mod N \tag{160}$$

Therefore, we have shown that pf-$\mathrm{TY}^{t,\kappa,\beta}_{\mathbb{Z}_N}$, the parafermionic condensation of $\mathrm{TY}^{t,\kappa}_{\mathbb{Z}_N}$ category, is uniquely determined by the data $t$, $\kappa$ and half-braiding $\beta$. For example, in the case of $N=2$, i.e. the fermionic condensation of $\mathrm{TY}(\mathbb{Z}_2)$, we have $t=1$, $\kappa=\pm1$ and $\beta=\pm i$ as in (137). Therefore, overall there are 4 gauge independent solutions that have been found in [61].

## Acknowledgements

We would like to thank Yongchao Lü who mentioned this interesting topic to us, and also Chi-Ming Chang, Zhuo Chen, Wei Cui, Zhihao Duan, Zheng-Cheng Gu, Yongchao Lü, Jia Qiang, and Fengjun Xu for their helpful discussions and related projects.

**Funding information**    The work of J.C. is supported by the Fundamental Research Funds for the Central Universities (No.20720230010) of China. B.H. is supported by the Young Thousand Talents grant of China as well as by the NSFC grant 12250610187. Q.-R.W. is supported by the National Natural Science Foundation of China (Grant No. 12274250).

## A   Pre-metric group $\mathcal{C}(G, q)$

In this appendix, we will introduce the concept of the pre-metric group $\mathcal{C}(G, q)$, associated with an Abelian group $G$ that forms a pre-modular category. The simple object of $\mathcal{C}(G, q)$ is labeled by an element of the group $G$. A quadratic form $q : G \to \mathbb{C}$ is a function such that $q(g) = q(g^{-1})$ and the function

$$b(g, h) := \frac{q(gh)}{q(g)q(h)} \tag{161}$$

is a bicharacter, i.e., $b(g_1 g_2, h) = b(g_1, h)b(g_2, h)$. The pre-metric group is related to the topological orders of Abelian anyons, wherein the quadratic form $q(g)$ plays the role of the self-half-braiding $\theta_g$ of the anyons.

    Form now on, we will focus on the case of $G = \mathbb{Z}_N = \{0, 1, ..., N-1\}$. The pre-metric group $\mathcal{C}(\mathbb{Z}_N, k) = \mathcal{C}(\mathbb{Z}_N, q_k)$ is characterized by an integer parameter $k$ with quadratic form $q_k$ defined as:

$$q_k(n) = (\omega_{N,k})^{n^2} = \begin{cases} e^{2\pi i \frac{k}{2N} n^2}(k = 0, 1, ..., 2N-1), & \text{if } N \text{ is even} \\ e^{2\pi i \frac{k}{N} n^2}(k = 0, 1, ..., N-1), & \text{if } N \text{ is odd} \end{cases}, \tag{162}$$

where for simplicity we introduced the phase factor

$$\omega_{N,k} := e^{2\pi i \frac{k}{\gcd(2,N)N}}. \tag{163}$$

In particular, the generator of $\mathbb{Z}_N$ has self-statistics

$$q_k(1) = \omega_{N,k}, \tag{164}$$

or topological spin

$$h_1 = \frac{k}{\gcd(2, N)N}. \tag{165}$$

From this expression, we can determine the parameter $k$ for $\mathcal{C}(\mathbb{Z}_N, q_k)$ based on the conformal spins of primary fields in a CFT. This allows us to identify the corresponding pre-metric group for the $\mathbb{Z}_N$ subcategory of primary fields in the CFT.

    To comprehend the conditions under which the $\mathbb{Z}_N$ symmetry is non-anomalous and amenable to gauging or orbifolding, we must examine the associator or $F$ symbol of the fusion category $\mathcal{C}(\mathbb{Z}_N, q_k)$. Notably, we can demonstrate that the $F$ symbols are as follows:

$$F^{a,b,c} = \begin{cases} (-1)^{ka\lfloor \frac{b+c}{N} \rfloor}, & \text{if } N \text{ is even} \\ 1, & \text{if } N \text{ is odd} \end{cases}. \tag{166}$$

From this expression, we can observe that the conditions for the $\mathbb{Z}_N$ symmetry to be anomalous or non-anomalous ($F = 1$) are:

$$\mathbb{Z}_N \text{ anomalous} \iff N \text{ is even and } k \text{ is odd}, \tag{167}$$

$$\mathbb{Z}_N \text{ non-anomalous} \iff N \text{ is odd or } k \text{ is even}. \tag{168}$$

    The category $\mathcal{C}(\mathbb{Z}_N, q_k)$ is not only fusion but also braided. The quadratic form of the pre-metric group is related directly to the twist of the braided tensor category

$$\theta_n = R^{n,n} = q_k(n) = (\omega_{N,k})^{n^2}, \tag{169}$$

as stated before. The braiding data $R$ is given by

$$R^{m,n} = (\omega_{N,k})^{mn}. \tag{170}$$

As a result, the pre-metric group $\mathcal{C}(\mathbb{Z}_N, q_k)$ is in fact a pre-modular category with both fusion and braiding structures.

One can also derive the modular matrices of the pre-modular category $\mathcal{C}(\mathbb{Z}_N, q_k)$ from the general relation

$$S_{\alpha\beta} = \frac{1}{D} \sum_\gamma N_{\alpha\beta}^\gamma \frac{\theta_\gamma}{\theta_\alpha \theta_\beta} d_\gamma = S_{\beta\alpha}, \quad D = \sqrt{\sum_\alpha d_\alpha^2}, \tag{171}$$

$$T_{\alpha\beta} = \theta_\alpha \delta_{\alpha\beta}. \tag{172}$$

The final results are

$$S_{m,n} = \frac{1}{\sqrt{N}} \frac{q_k(m+n)}{q_k(m)q_k(n)} = \frac{1}{\sqrt{N}} (\omega_{N,k})^{2mn} = \begin{cases} \frac{1}{\sqrt{N}} e^{2\pi i \frac{k}{N} mn}, & \text{if } N \text{ is even} \\ \frac{1}{\sqrt{N}} e^{2\pi i \frac{k}{N} 2mn}, & \text{if } N \text{ is odd} \end{cases}, \tag{173}$$

$$T_{m,n} = q_k(n)\delta_{mn}. \tag{174}$$

The physical meaning of the bicharacter Eq. (161) is in fact the monodromy matrix as

$$b(m,n) = \frac{q(m+n)}{q(m)q(n)} = M_{m,n} = \frac{S_{mn}S_{00}}{S_{0m}S_{0n}} = (\omega_{N,k})^{2mn}. \tag{175}$$

From the modular matrices, one can also determine when the pre-metric group (or pre-modular category) $\mathcal{C}(\mathbb{Z}_N, q_k)$ is metric (or modular). In fact, we have the condition

$$\mathcal{C}(\mathbb{Z}_N, k) \text{ is modular} \iff q_k \text{ is non-degenerate} \iff \gcd(k, N) = 1. \tag{176}$$

In practice, the structure of the pre-metric group $\mathcal{C}(\mathbb{Z}_N, q_k)$ provides us with valuable insights into the properties of the $\mathbb{Z}_N$ subcategory of primary fields within a CFT. For example, when we have the conformal dimension $h_1$ of the $\mathbb{Z}_N$ primary field generator, Eq. (165) comes into play, allowing us to deduce the parameter $k$. This parameter uniquely defines the pre-metric group or pre-modular category $\mathcal{C}(\mathbb{Z}_N, q_k)$. Once we have determined the value of $k$, Eq. (168) becomes instrumental in identifying when the $\mathbb{Z}_N$ symmetry is non-anomalous. It is crucial in deciding whether we can engage in process of orbifolding, gauging, or condensing this symmetry. Furthermore, the condition given by Eq. (176) aids in uncovering the structure of the primary field category within a CFT. If the $\mathcal{C}(Z_N, q_k)$ subcategory also holds modularity, then the entire category can be represented as the Deligne product of $\mathcal{C}(\mathbb{Z}_N, q_k)$ and another subcategory.

# B  $\mathbb{Z}_N$-graded division algebra and $q_M$-type simple object in para-fusion category

In a fusion category, the endomorphism algebra of any simple object must be a division algebra according to Schur's lemma. Similarly, if $X$ is a simple object in a $\mathbb{Z}_N$-graded fusion category (or para-fusion category) $\mathcal{C}$, then $\text{End}(X) = \text{Hom}_{\mathcal{C}}(X, X)$ should be a $\mathbb{Z}_N$-graded division algebra.

In the context of a fusion category, a simple object $X$ is associated with the endomorphism algebra $\text{End}(X) = \mathbb{C}$, as the only division algebra over $\mathbb{C}$ is itself. However, the situation changes when dealing with more complicated structures, such as $\mathbb{Z}_2$-graded fusion categories or super-fusion categories. In these cases, the endomorphism algebra $\text{End}(X)$ can expand to

include the two-dimensional Clifford algebra $\mathbb{C}l_1 = \mathbb{C}^{1|1}$, which contains both even and odd generators. To gain a deeper understanding of the simple objects within para-fusion categories, it becomes crucial to explore the classification of $\mathbb{Z}_N$-graded division algebras .

The classification theorem is as follows [98, 99]:

**Theorem 1.** *Let $R = \oplus_{g \in \mathbb{Z}_N} R_g$ be a finite-dimensional $\mathbb{Z}_N$-graded algebra over $\mathbb{C}$. Then $R$ is a graded division algebra if and only if $R$ is isomorphic to the group algebra $\mathbb{C}[\mathbb{Z}_M] = \oplus_{h \in \mathbb{Z}_M} \mathbb{C}_h$, where $\mathbb{Z}_M$ is a finite subgroup of $\mathbb{Z}_N$, i.e., $M$ is a divisor of $N$.*

Consequently, in a $\mathbb{Z}_N$ para-fusion category, there may exist several distinct types of $q$-type objects. The count of these types is determined by the divisor function $\tau(N)$, which represents the number of divisors of $N$. For instance, a few examples of $\tau(N)$ for small values of $N$ are: $\tau(N) = 1, 2, 2, 3, 2, 4, 2, 4, 3, 4$ for $1 \leq N \leq 10$.

For instance, let's consider the case of $N = 2$. In this scenario, there are $\tau(2) = 2$ subgroups of $\mathbb{Z}_2$, leading to two distinct types of $\text{End}(X)$ for simple object $X$. The algebra $\mathbb{C}[\mathbb{Z}_1] = \mathbb{C}$ corresponds to the $m$-type simple object, while $\mathbb{C}[\mathbb{Z}_2] = \mathbb{C}^{1|1} = \mathbb{C}l_1$ (the first complex Clifford algebra) corresponds to the $q$-type simple object.

If $\text{End}(X) = \mathbb{C}[\mathbb{Z}_M]$ within a $\mathbb{Z}_N$-graded fusion category, we will say $X$ to be a $q_M$-type object. It's worth noting that when $M = 1$, $q_1$-type is equivalent to what we refer to as an $m$-type object in super-fusion category.

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
