# Peer review of "Para-fusion Category and Topological Defect Lines in $\mathbb Z_N$-parafermionic CFTs"

_SciPost Physics_

## Round 1 · Referee Report · Anonymous (Referee 1) · 2024-2-17

Strengths

The paper attempts to fill the clear desire for a theory of parafermionization, analogous to theories of fermionization that have been developed in the preceding years. This is both an intellectually satisfying pursuit, but also a potentially useful avenue given the value of fermionization in understanding anomalies, non-invertible defects, and solving explicit models, among other things.

The authors identify clear and convincing targets for such a parafermionic program:
-- Consistency with understanding of parafermionic models in 2d CFTs
-- Definition of para-fusion categories, analogous to super-fusion categories in the $\mathbb{Z}_2$ case
-- A general theory of parafermionic anyon condensation
-- Relations to anomalies and $\mathbb{Z}_n$ duality defects (Tambara-Yamagami categories)

The authors also provide reviews of the literature:
-- A review of a general theory of parafermionization (beyond just 2d CFTs) proposed in previous work
-- Reviews of anyon condensation in the bosonic and fermionic case

The paper is also well-formatted.

Weaknesses

With these positives in mind, I have concerns ranging from minor to major with the draft.

Major: Although the authors cannot define para-spin structures in 2d, this is fine if they are defined in some future work in the parafermionization community. However, I am not convinced that there exists a consistent unitary invertible field theory called para-Arf. In particular, an important claim is that there exists a "parafermionic chain" (defined in a reference) whose low energy partition function is $\omega^{\mathrm{Arf}_N(\rho+s)}$, which on the torus is given by $\mathrm{Arf}_N(\rho) = \rho_1 \rho_2$.

In eqn (26) the authors define bosonization and parafermionization, and it is easy to confirm that these maps are inverses with the prescriptions given. A first question: Why do we have this prescription for bosonization and parafermionization? Why can we not stack with many copies of $\mathrm{Arf}_N$ before summing over spin-structures or background connections? E.g. for $N=5$, why can I not sum over background connections with $\omega^{m\mathrm{Arf}_5[\rho+s]}$ for $m=2,3,4$? Perhaps different $m$ give different parafermionizations, but it is not commented on at all.

However, using the parafermionization/bosonization prescriptions in (26), I still find issues if one believes the para-Arf theory actually exists as an invertible topological phase. Consider the following (say $N$ is prime for simplicity): 1. Start with a modular invariant bosonic theory $T$. 2. Parafermionize to $T_{pf}$. 3. Stack with $k$ copies of para-Arf, $T_{pf} + k \mathrm{Arf}_n$ 4. Re-bosonize to the bosonic theory $T'$ It appears that, unless $N=2$ or $k=0$, the resulting theory $T'$ is not modular invariant under a modular T-transformation. So stacking with any number of para-Arfs (besides 0) on the parafermionic side seems to take us out of the space of consistent bosonic theories.

In the paper, it is noted that to obtain the (consistent) bosonic theory $T/\mathbb{Z}_N$, one has to stack the parafermionic theory $T_{pf}$ with para-Arf and then perform a "conjugation" before bosonizing. This conjugation seems mysterious and unexplained. But one can take its existence for granted as something that requires further investigation. In this case $N$ is prime, charge-conjugation and stacking with para-Arf seem to give a $D_{2N}$ of operations on the parafermionic theory. Still the same modular invariance inconsistencies arise from before; except in the case with no charge conjugation and no para-Arf, or the case of charge conjugation and 1 para-Arf.

In any case, I am not currently convinced that the para-Arf phase exists as a unitary invertible phase and follows the properties one expects of an invertible phase.

Middle: -- There are concerning sloppy comments made (especially in the introduction) which tinge the rest of the paper. Here are 3 early examples and 1 later example: 1. "Historically, these defect line objects were investigated because of their connections to boundary CFTs, twisted partition functions, orbifolds, and associated SymTFTs." I am slightly confused what the "associated SymTFT" is. There are many SymTFTs that can be associated to a theory, there is rarely ever a "maximal" one. e.g. what is the associated SymTFT to a compact boson? 2. "The collections of non-invertible TDLs together with ordinary invertible ones, for a given 2d CFT or TFT, are described in the mathematical language of the fusion category" Not all TDLs will form a fusion category, maybe some subset will. For example, a fusion category must have finitely many isomorphism classes of simple objects. This is not the case, again, for the TDLs of the compact boson CFT. 3. "2d TDLs can be naturally interpreted as anyons in three dimensions" this is true and gives helpful insight into the work the authors pursue, but it is a little misleading to say that anyons are lifted into the bulk. Many bulk anyons can condense into 1 on the boundary, or bulk anyons may split into many on the boundary. 4. More generally, later on in the paper I find it confusing that the authors talk about condensing anyons in fusion categories that do not appear to be braided. --In example 4.1 the authors talk about the 3-state Potts model, then present the partition function (and discuss the model) as the diagonal modular invariant for $W_3$ chiral algebra, not the minimal model. -- There is far too much review. E.g. pages 14-25 are about anyon condensation, but only pages 23-25 are about the novel para-fermionic case. -- I could not find the outright definition of a parafermionic line in a UMTC. For example, in a Z3-parafermionic scenario, is it a line with spin-1/3 only? Based on their example 4.1, it seems a parafermionic line can have spin 2/3. Is the condition that it have spin congruent to 0 mod 1/3, but not be exactly 0, and generate a Z3 sub fusion category?

Minor: -- There are minor grammar and spelling mistakes. Especially in the introduction. E.g. "Fermoinic" or inconsistency with "spatial" vs "spacial" etc. which I will not continue to list. The reader familiar with the subject can resolve them, but they are distracting. -- There are some phrases whose purpose are confusing, redundant, or sound logically out of order. As 2 random examples early on: 1. "On the other hand, for the latter non-invertible symmetries, they are referred to as topological surface operators that are not invertible." I understand that this sentence is trying to say "non-invertible symmetries correspond to topological surface operators that do not close under fusion" or something akin to that. But the phrasing is strange and would not be useful e.g. to a grad student or newcomer to the topic reading the introduction. 2. The paragraphs reviewing "orbifolding" before (2) appear to be jumbled and out of order. Paragraph 1 identifies orbifolding with gauging and discusses the relation to twisted boundaries by TDLs, while paragraph 2 seems to explain the background again in a jumbled order. -- Additional references on fermionic CFTs and minimal models include: "Fermionic CFTs and classifying algebras" by Runkel and Watts; very early work by Petkova et al. "Two-dimensional (Half) Integer Spin Conformal Theories With Central Charge c < 1" and "Fusion Matrices and c < 1 (Quasi)local Conformal Theories"; or follow-up "Boundary States and Anomalous Symmetries of Fermionic Minimal Models" by Boyle-Smith. The first 3 papers are earlier than the works cited, so especially deserve some recognition.

Report

I do not believe the journal's acceptance criteria are presently met, and significant conceptual work will be required to convince the reader that the work is on solid footing.

Requested changes

My requested major changes are:
-- Justify the consistency of the para-Arf phase as an invertible phase or explain the bosonization and parafermionization procedure in a different self-consistent way
-- I would also hope to see big improvements in the precision of the language and highlighting of novel work. Highlighting of novel work could be done explicitly with phrases or boxes, or implicitly, by putting significant extended reviews into appendices.

---

## Editorial Decision

awaiting_resubmission